# Large Language Model Agents Are Not Always Faithful Self-Evolvers

**Weixiang Zhao**[1]   **Yingshuo Wang**[1]   **Yichen Zhang**[1]   **Yang Deng**[2]
**Yanyan Zhao**[1]   **Wanxiang Che**[1]   **Bing Qin**[1]   **Ting Liu**[1]

## Abstract

Self-evolving large language model (LLM) agents continually improve by accumulating and reusing past experience, yet it remains unclear whether they faithfully rely on that experience to guide their behavior. We present the first systematic investigation of *experience faithfulness*—the causal dependence of an agent's decisions on the experience it is given—in self-evolving LLM agents. Using controlled causal interventions on both raw and condensed forms of experience, we comprehensively evaluate four representative frameworks across 13 LLM backbones and 9 environments. Our analysis uncovers a striking asymmetry: while agents consistently depend on raw experience, they often disregard or misinterpret condensed experience, even when it is the only experience provided. This gap persists across single- and multi-agent configurations and across backbone scales. We trace its underlying causes to three factors: the semantic limitations of condensed content, internal processing biases that suppress experience, and task regimes where pretrained priors already suffice. These findings challenge prevailing assumptions about self-evolving methods and underscore the need for more faithful and reliable approaches to experience integration. Our code is available at: https://github.com/Dreamcatcher0622/Faithfulness.

## 1. Introduction

The emergence of self-evolving agents represents a pivotal step in the development of autonomous systems capable of continuous learning and adaptation (Zhao et al., 2024b; Dou et al., 2025; Silver & Sutton, 2025). Unlike the traditional static paradigms, these agents dynamically gather, store and reuse experiences from their interactions with the environment to inform future decisions (Gao et al., 2025; Cai et al., 2025; Bell et al., 2025; Hendrycks et al., 2025).

At the center of this paradigm is the use of experience. Such experience generally falls into two categories: raw and condensed (Hu et al., 2025; Zhang et al., 2025b). As demonstrated in the left part of Figure 1, raw experiences capture concrete historical traces, such as successful trajectories from similar tasks, that agents can directly reference or replay (Zhao et al., 2024a; Zhang et al., 2025a). Condensed experiences, by contrast, are distilled from those traces and encode transferable insights, including abstract plans or failure heuristics (Ouyang et al., 2025; Wang et al., 2025). Despite their central role, prior work has focused mainly on how such experiences are stored or represented, leaving it unclear whether agents actually and faithfully leverage them to improve performance. To address this, we present the first systematic investigation into the **faithfulness** of experience utilization in self-evolving LLM agents, organized around two core research questions (RQs).

We begin by systematically examining (**RQ1**) *is the performance improvement of self-evolving agents faithfully attributable to their use of past experiences?* (§3 & §4). To answer this, we introduce a suite of controlled causal interventions targeting both raw and condensed experiences, and assess how such perturbations affect downstream behavior. To illustrate this, Figure 1 shows a motivating example where raw and condensed experiences are perturbed in different ways. We define *experience faithfulness* as the extent to which an agent's behavior is causally grounded in its input experience—i.e., if perturbing the experience leads to significant behavioral changes, we consider the agent to have faithfully used it. Our evaluation spans four representative self-evolving frameworks, encompassing both offline (Zhao et al., 2024a) and online (Ouyang et al., 2025) paradigms, across single-agent and multi-agent settings (Zhang et al., 2025a). We benchmark 13 diverse LLM backbones across 9 environments, including reasoning, web interaction, and embodied decision-making, providing comprehensive coverage of both model families and application settings.

We first show that agents are consistently more faithful

---

[1]Harbin Institute of Technology, China [2]Singapore Management University, Singapore. Correspondence to: Yanyan Zhao <yyzhao@ir.hit.edu.cn>.

*Proceedings of the 43rd International Conference on Machine Learning*, Seoul, South Korea. PMLR 306, 2026. Copyright 2026 by the author(s).

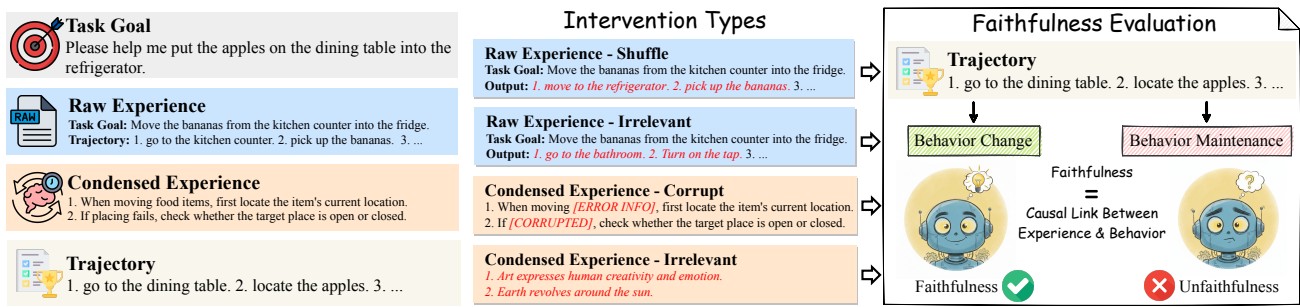

*Figure 1.* Examples of experience intervention and faithfulness evaluation. Given a task goal, the agent receives *raw experience* (concrete historical trajectories that succeed to complete the similar tasks) and *condensed experience* (abstract summaries or heuristics). We apply different types of interventions, such as shuffling, corrupting, or replacing experience with irrelevant content, to test whether such perturbations affect downstream behavior. A full taxonomy of intervention types is provided in Section 3. Faithfulness is determined by whether the agent's behavior causally changes in response to the perturbed input.

to raw experiences than to condensed ones when both are present, exhibiting substantial behavioral changes under raw experience perturbations but not under condensed ones (§4.1). We further demonstrate that this lack of faithfulness to condensed inputs persists even when raw experience is entirely absent, indicating that the problem is not due to competition or overshadowing (§4.2). Extending our analysis to collaborative multi-agent settings, this asymmetry remains: agents reliably exploit raw trajectories while largely ignoring the semantic content of condensed summaries (§4.3). Finally, this faithfulness disparity proves robust across model scales: while larger models achieve higher overall performance, they still fail to meaningfully ground their behavior in condensed experience (§4.4). These findings reveal a core limitation of current self-evolving agents: although they benefit from accumulated experience, they nonetheless display pronounced faithfulness failures—most notably in how they utilize condensed experience.

These findings naturally lead to our second question: (**RQ2**) *why do self-evolving agents often fail to faithfully leverage condensed experiences?* (§5) We trace this to a cascading triad of causes rooted in the three core components of self-evolving systems. First, condensed experiences themselves are often semantically limited—many encode only vague heuristics or generic summaries, lacking the specificity required to guide behavior (§5.1). Second, even when relevant content is present, agents often fail to utilize it due to internal processing biases (Mohsin et al., 2025) that favors local contextual signals over retrieved information (§5.2). Finally, the structure of the task further compounds this issue: for certain types such as knowledge-intensive benchmarks, agents often succeed by relying solely on their pretrained semantic priors (Shi et al., 2024), reducing the marginal utility of retrieved experience and diminishing the model's incentive to incorporate external guidance at all (§5.3).

In summary, our findings challenge the common assumption that self-evolving agents faithfully leverage their accumu-

lated experiences. Despite performance gains, agents often ignore or misuse condensed experience, revealing a significant gap between utility and faithfulness. Our study provides a principled framework to diagnose this issue and underscores the need for more reliable and interpretable mechanisms for experience-driven adaptation in LLM agents.

## 2. Preliminaries

We define *self-evolving agents* as agents that progressively improve their behavior by *accumulating*, *retrieving*, and *exploiting* past experiences, without modifying the underlying model parameters (Gao et al., 2025).

After each interaction with the environment, the agent produces a trajectory $\tau$ and receives feedback $r$. From this $(\tau, r)$ pair, the system may store two types of experience:

**Raw Experience** $E^{\text{raw}}$: detailed traces of observations, actions, intermediate states, and rewards from successful trajectories that the agent can directly reference or replay.

**Condensed Experience** $E^{\text{cond}}$: high-level summaries (e.g., heuristics or causal lessons) distilled from both successful and failed trajectories, capturing generalizable structure.

All accumulated experiences are stored in a shared external repository $M = \{E_1, E_2, \ldots, E_n\}$, where each $E_i$ is either a raw experience $E_i^{\text{raw}}$ or a condensed experience $E_i^{\text{cond}}$.

At inference time, given a new task input $x$, the agent retrieves a relevant subset of experiences $M(x) \subset M$, which may contain $M^{\text{raw}}(x)$, $M^{\text{cond}}(x)$ or both, depending on the framework design. The retrieved experiences are used to augment the input and yield the output $y$.

$$y = \pi_\theta\big([x; M^{\text{raw}}(x); M^{\text{cond}}(x)]\big),$$

We consider both *offline* self-evolving settings, where $M$ is fixed, and *online* self-evolving settings, where $M$ evolves dynamically with ongoing interactions.

# 3. Causal Intervention on Experience

To assess whether agents faithfully exploit retrieved experience during inference, we design controlled interventions that selectively perturb the raw or condensed experience.

## 3.1. Experimental Setup

**Agent Framework.** We evaluate four representative self-evolving agents. In the *offline single-agent* setting, we use **ExpeL** (Zhao et al., 2024a); for *online single-agent* settings, we include **Dynamic Cheatsheet** (Suzgun et al., 2025) and **ReasoningBank** (Ouyang et al., 2025); and for *online multi-agent*, we use **G-Memory** (Zhang et al., 2025a). In all frameworks, the LLM backbone remains frozen, and behavioral adaptation arises solely from the accumulation, retrieval, and exploitation of external experiences. Further details are provided in Appendix A.

**Backbone Model.** Our experiments span 10 LLMs, including closed-source models: **GPT-4o(-mini)** (Hurst et al., 2024), **GPT-5.2** (Singh et al., 2025), **Gemini-2.5-Flash** (Comanici et al., 2025), **Gemini-3-Pro** (DeepMind, 2025) and **Claude-Sonnet-4.6**, which largely follow the official settings adopted in their respective agent frameworks. In addition, we include a range of open-weight **Qwen3** (Yang et al., 2025a) variants, 1.7B–32B dense models, 30B-A3B, and 235B-A22B MoEs, to enable more systematic analysis across model scales and architectures.

**Environment & Benchmark.** Follow the official setting in each agent framework, we evaluate across 9 benchmarks in 4 domains: (1) For *knowledge-intensive question answering*, we include **HotpotQA** (Yang et al., 2018), **FEVER** (Thorne et al., 2018), **GPQA-Diamond** (Rein et al., 2024), and **MMLU-Pro Eng.** (Wang et al., 2024). (2) For *mathematical reasoning*, we use **AIME 2024** and **Game of 24** (Yao et al., 2023a; Suzgun & Kalai, 2024). (3) For *embodied action*, we adopt the interactive environment **ALFWorld** (Shridhar et al., 2021). (4) For *web interaction*, we evaluate on **WebArena** (Zhou et al., 2024) and **WebShop** (Yao et al., 2022). These provide a diverse testbed for experience-based adaptation (details in Appendix B).

**Implementation Details.** Closed-source and large-scale open-weight models are accessed via official APIs, while other open-weight models are deployed locally with vLLM (Kwon et al., 2023) on NVIDIA A800 GPUs. We strictly follow the official configuration of each agent framework. Additional details are provided in Appendix C.

## 3.2. Raw Experience Interventions

Raw experiences typically consist of full interaction trajectories, including observations, thoughts, and actions (Yao et al., 2023b). Since these experiences preserve fine-grained behavioral sequences, our interventions aim to test whether agents rely on their temporal structure, semantic relevance, or simply their presence to guide decisions. We introduce three types of interventions designed to probe different aspects of raw experience utilization:

**Empty**: Remove all semantic content from retrieved raw experiences, while retaining their formatting cues (e.g., prompts like "Here are two examples of successful trajectories:"). This differs from a simple ablation (`w/o raw`), which omits the experience section entirely.

**Shuffle**: Randomly shuffle the order of steps within each trajectory, preserving content tokens but disrupting temporal coherence and causal structure.

**Irrelevant**: Replace retrieved trajectories with ones sampled from other unrelated tasks, preserving format and structure but removing topical and semantic relevance.

## 3.3. Condensed Experience Interventions

Condensed experiences consist of distilled summaries. Unlike raw trajectories, they encode high-level abstractions such as heuristics or causal lessons. To probe the faithfulness, we introduce four types of interventions:

**Empty**: Remove all semantic content from the condensed experience while preserving the formatting cues (e.g., "Here is a distilled insight from past trajectories:" followed by an empty slot). This is distinct from a full ablation (`w/o cond`), which omits the condensed experience entirely.

**Corrupt**: Randomly alter key components (e.g., distorting action references) to break internal coherence.

**Irrelevant**: Replace the condensed summary with one that is entirely unrelated to the current task goal, using a generic and task-agnostic description.

**Filler**: Replace the entire content of condensed experience with semantically empty placeholder tokens (e.g., special characters such as "%$#&"), preserving surface structure while removing all meaningful information.

These interventions allow us to test whether performance gains arise from the actual semantics of condensed experience. More detailed examples and the rationale behind these intervention designs can be found in Appendix D.

# 4. Evaluation of Experience Faithfulness

## 4.1. Faithfulness under Joint Raw & Condensed Access

We begin by examining settings where both raw and condensed experiences are simultaneously provided to the agent. Figure 2 reports results in the offline ExpeL framework, while Figure 3 shows online performance using the Dynamic CheatSheet setup. To further validate the generality of our

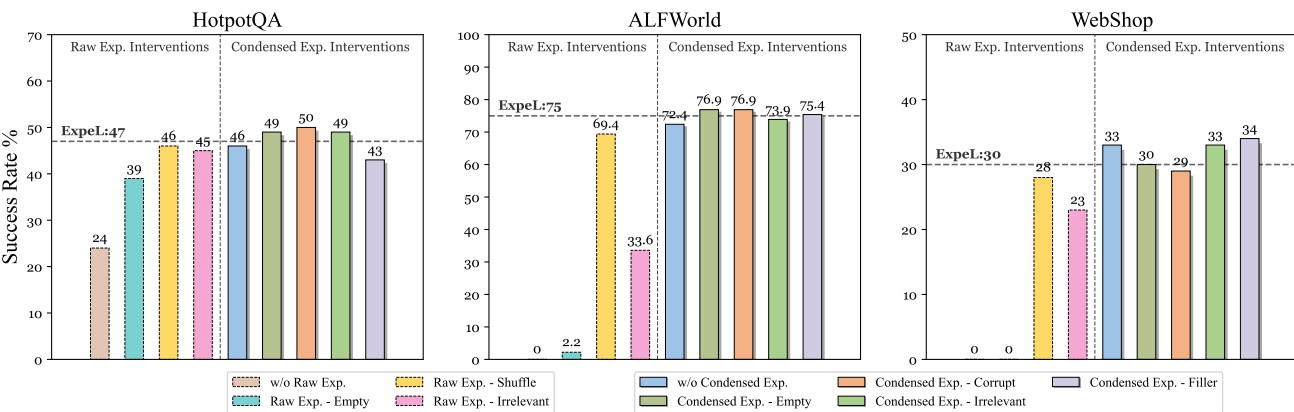

*Figure 2.* Intervention results on the ExpeL framework (offline, single-agent) using GPT-4o across three benchmarks. ExpeL consistently relies more on raw trajectories, while showing weak or inconsistent sensitivity to condensed summaries.

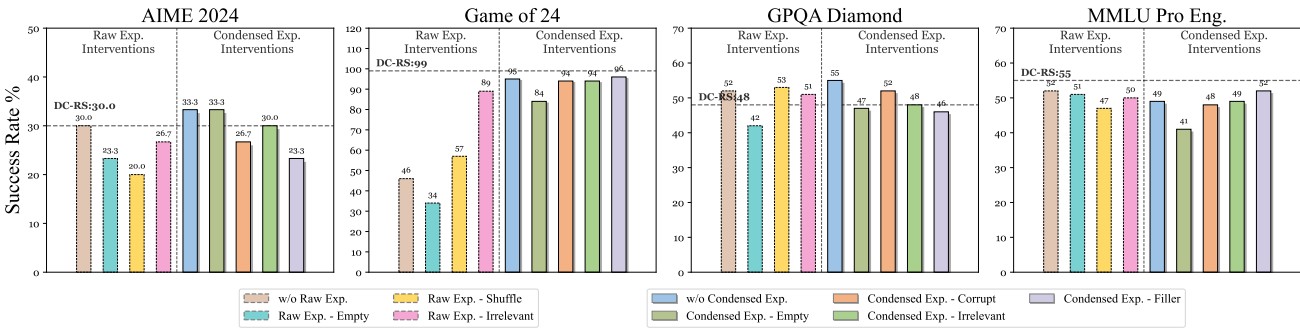

*Figure 3.* Intervention results on the Dynamic CheatSheet (DC-RS) framework (online, single-agent) using GPT-4o. Raw experience perturbations significantly reduce performance, whereas condensed experience manipulations often have negligible impact.

findings, we additionally replicate the same analysis on stronger models, including Qwen3-235B-A22B and Claude-Sonnet-4.6 (Appendix D.3). Across both open-weight and frontier closed-source models, we observe highly consistent patterns, affirming that the raw–condensed faithfulness asymmetry is not restricted to a particular model family. This setting provides our first set of faithfulness insights.

**Faithfulness to raw experience is strong and robust.** Across both frameworks, we find that removing raw experience (`w/o Raw Exp.`) reliably causes substantial performance degradation on most tasks, demonstrating that raw experience is indeed a primary contributor to performance. Perturbing raw trajectories, particularly through `Empty` or `Irrelevant` replacements, produces similarly severe declines. These results indicate that agents truly leverage the semantic and temporal structure encoded in raw trajectories.

**Condensed experience often has minimal behavioral influence.** In contrast, most interventions on condensed experience lead to little or no change in behavior. Across both frameworks and nearly all tasks, perturbations such as `Corrupt`, `Irrelevant`, and `Filler` yield performance that is nearly indistinguishable from the unperturbed baseline. Likewise, even removing condensed experience

altogether (`w/o Condensed`) has only marginal impact. These patterns suggest that agents either struggle to interpret condensed summaries or simply do not rely on them during decision-making—despite their explicit presence in the input. Taken together, this reveals a serious faithfulness gap in how agents purportedly "use" condensed experience.

**Consistent patterns across offline and online paradigms.** Notably, these phenomena hold across both offline (ExpeL) and online (Dynamic CheatSheet) self-evolving paradigms. Despite their different mechanisms for experience accumulation, the resulting faithfulness patterns remain strikingly similar. In both cases, agents strongly depend on raw trajectories while exhibiting weak or inconsistent reliance on condensed summaries. This consistency across paradigms underscores that the faithfulness gap is a fundamental property of current self-evolving designs, rather than an artifact of a specific memory-update strategy.

**Task-specific characteristics modulate experience sensitivity.** We observe that in certain knowledge-intensive tasks such as GPQA-Diamond and MMLU-Pro Eng., the agent exhibits comparable sensitivity to both raw and condensed experience. This suggests that task type or structure may affect how different forms of experience are utilized, a

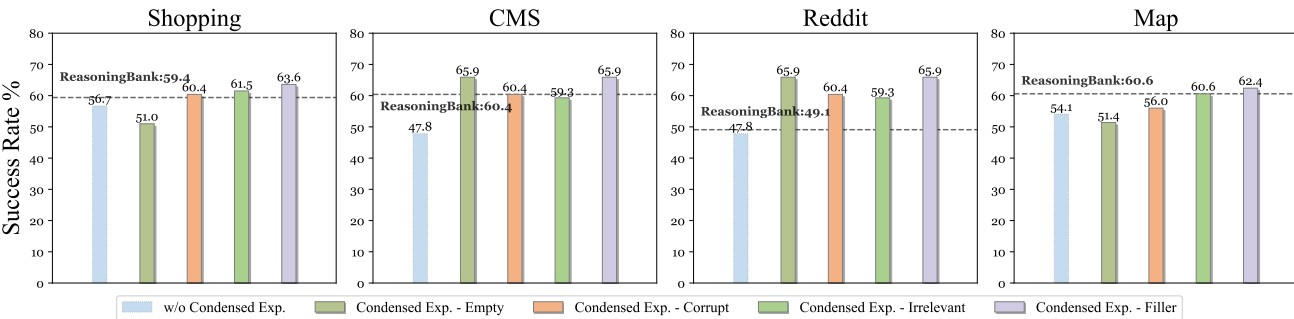

*Figure 4.* Impact of condensed experience interventions on the ReasoningBank framework (online, single-agent) using Gemini-2.5-Flash across four WebArena sub-tasks. Despite the absence of raw experience, agents show only mild sensitivity to semantic manipulations of condensed experience, indicating limited semantic faithfulness.

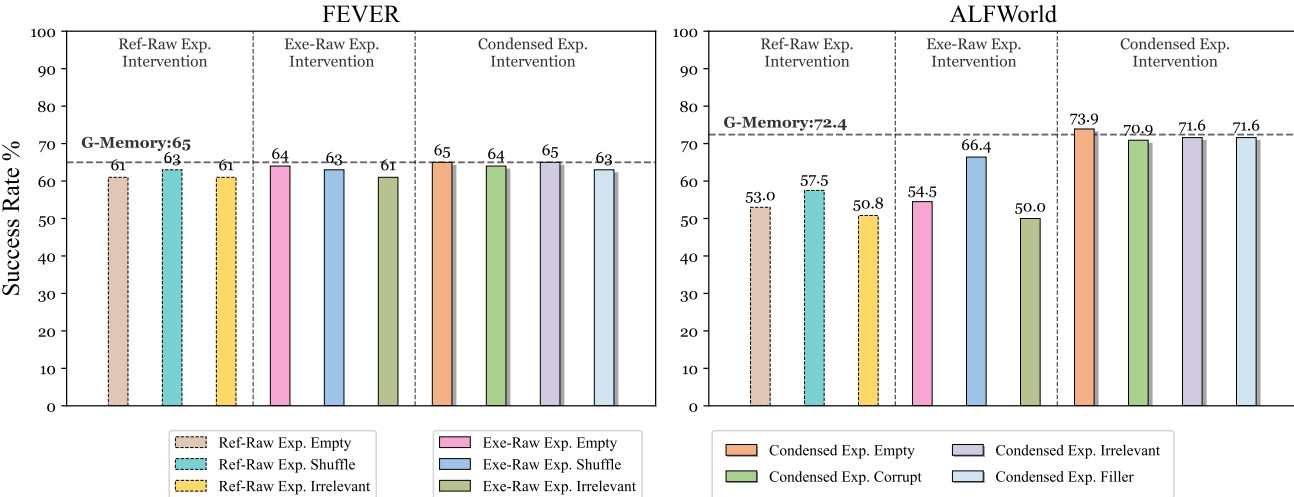

*Figure 5.* Faithfulness interventions on the G-Memory framework (online, multi-agent) with GPT-4o-mini. Agents access two forms of raw experience (reference and execution) and one form of condensed experience.

phenomenon we discuss further in §5.3.

## 4.2. Faithfulness under Condensed-Only Input

Given that agents exhibit stronger faithfulness to raw experiences than condensed ones, a natural follow-up question arises: *Is this lack of faithfulness to condensed experiences due to the presence of raw experience overshadowing it?* To isolate this effect, we examine the ReasoningBank framework, which provides *only condensed experience* and contains no raw trajectories. This setting enables a clean assessment of whether agents meaningfully rely on condensed summaries when no richer experience is available.

**Condensed experience improves performance, but not through faithful utilization.** As shown in Figure 4, removing condensed experience (`w/o Condensed Exp.`) leads to consistent performance drops across all four WebArena tasks, indicating that even in the absence of raw trajectories, condensed summaries provide useful guidance that agents can leverage to improve their behavior.

However, this utility does not imply faithful grounding. The agent's responses remain surprisingly insensitive to semantic perturbations: interventions such as `Corrupt`, `Irrelevant`, and even `Filler` lead to only negligible degradation or even slight improvements.

These results suggest that while condensed experience does contribute to task success, the agent does not rely on its actual content in a faithful manner. Instead, performance gains may stem from superficial features, such as the presence of a text block or stylistic patterns (`Empty`), rather than genuine semantic grounding. We delve deeper into this phenomenon and its underlying causes in §5.

**Similar trends hold across other model families.** We further replicate this evaluation using Qwen3-14B, Qwen3-32B, and the frontier model Gemini-3-Pro on the same ReasoningBank setup. As detailed in Appendix D.4, we observe highly consistent patterns across all models: semantic perturbations on condensed experience still fail to consistently degrade success rates. This suggests that the limited

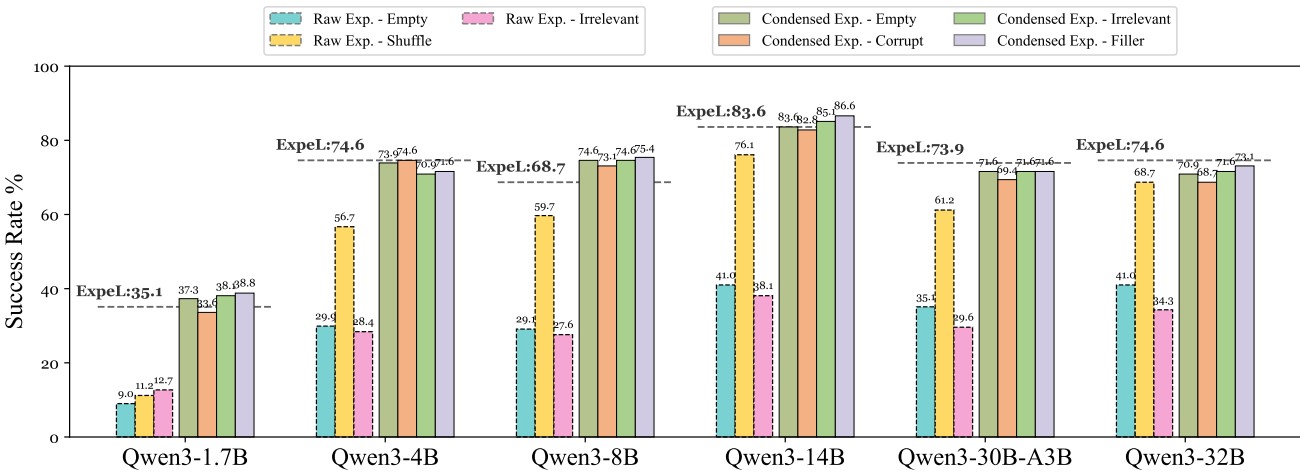

*Figure 6.* Model-scale-wise analysis of intervention sensitivity across six Qwen3 models (1.7B to 32B) with ExpeL.

faithfulness to condensed knowledge is a general behavioral tendency rather than a model-specific phenomenon.

### 4.3. Faithfulness under Multi-Agent Scenario

To further assess whether our conclusions hold in collaborative multi-agent settings, we evaluate the G-Memory framework. Results under the GPT-4o-mini backbone are shown in Figure 5. We additionally replicate the same evaluation on the stronger frontier model GPT-5.2, together with complementary experiments based on Qwen3-235B-A22B (Appendix D.5). Across all backbones, we observe highly consistent trends, further confirming that the raw–condensed faithfulness asymmetry persists in multi-agent settings.

**Raw experience from both sources is faithfully utilized.** G-Memory accumulates two types of raw experience: (1) *Reference Raw Experience (Ref-Raw Exp.)*, curated trajectories collected offline, and (2) *Execution Raw Experience (Exe-Raw Exp.)*, accumulated autonomously during agent operation. In Figure 5, we find that perturbing either type consistently results in obvious performance degradation on ALFWorld, confirming that both human-curated and self-collected raw experiences are faithfully utilized by the agent.

**Condensed experience remains fragile.** Consistent with our earlier findings, perturbations to the condensed experience yield only marginal effects. This further reinforces our central conclusion: although condensed experience can offer some utility, agents seldom anchor their decision-making process in its underlying semantics.

**Faithfulness inconsistencies persist in knowledge reasoning tasks.** However, in knowledge-grounded tasks such as FEVER, the impact of raw experience perturbation is less pronounced. This mirrors prior trends in GPQA-Diamond and MMLU-Pro Eng., where raw and condensed experience yield more comparable influence. We provide a detailed

diagnosis of these task-dependent effects in §5.3.

### 4.4. Faithfulness under Model Scaling

Finally, we ask whether scaling model size improves the degree to which agents faithfully rely on provided experience. Using the ExpeL framework, we conduct controlled interventions across six Qwen3 variants (from 1.7B to 32B parameters), shown in Figure 6.

**Larger models perform better, but the faithfulness gap remains.** As expected, scaling improves unperturbed success rates. Yet perturbation results show that, even at larger scales, models remain markedly more faithful to raw experience than to condensed representations.

**Scaling does not resolve condensed-experience unfaithfulness.** Notably, the condensed experience used in ExpeL is generated by the backbone model itself. Therefore, larger models should in principle produce higher-quality and more informative summaries. However, despite substantial performance improvements with scaling, both small and large models remain consistently insensitive to condensed experience interventions. This suggests that the observed unfaithfulness is not solely attributable to low-quality summaries, but reflects a more fundamental limitation in how current self-evolving agents utilize condensed experience.

These findings suggest that while parameter scaling enhances performance, it does not inherently resolve the core challenge of experience faithfulness.

## 5. The Cause of Unfaithfulness

In this section, we trace the cause of such counterintuitive unfaithfulness through the three foundational components of any self-evolving system: the experience itself (§5.1), the backbone model that processes it (§5.2), and the task

*Table 1.* Error distribution for ReasoningBank on WebArena when the agent succeeds without condensed experience but fails with it.

| | Distraction(%) | Reliance(%) | Premature(%) |
|---|---|---|---|
| *Gemini-2.5-Flash* | | | |
| Shopping | 45.2 | 32.3 | 22.6 |
| CMS | 40.9 | 27.3 | 31.8 |
| Reddit | 26.7 | 33.3 | 40.0 |
| Map | 18.8 | 31.3 | 50.0 |
| *Qwen3-32B* | | | |
| Shopping | 79.2 | 8.3 | 12.5 |
| CMS | 75.0 | 12.5 | 12.5 |
| Reddit | 61.9 | 9.5 | 28.6 |
| Map | 74.1 | 11.1 | 14.8 |
| *Qwen3-14B* | | | |
| Shopping | 74.3 | 11.4 | 14.3 |
| CMS | 86.7 | 6.7 | 6.7 |
| Reddit | 60.9 | 0 | 39.1 |
| Map | 72.7 | 9.1 | 18.2 |

environment in which the agent operates (§5.3).

### 5.1. Semantic Limitations of Condensed Experience

We first examine the role of the experience itself. A surprising observation from our interventions is that agents sometimes perform better when condensed experience is perturbed or removed. This suggests that condensed summaries may be uninformative or misaligned. To investigate this possibility, we analyze cases where the agent *succeeds without* condensed experience but *fails when* it is added. As shown in Table 1, most failures fall into three categories:

**Distraction from the task goal.** Condensed summaries can redirect the agent toward tasks implied by the retrieved heuristics rather than the actual user intent, leading to unnecessary detours or complete drift from the target. This is especially common for smaller models, where high-level heuristics easily override task grounding.

**Overreliance on incorrect priors.** Condensed experience often encodes assumptions about element types, layouts, or workflows that do not match the current state. Agents may rigidly follow these outdated patterns instead of inspecting the live page, resulting in invalid or misplaced actions.

**Premature inference from prior patterns.** Agents frequently jump to conclusions when summaries suggest that certain items "should" exist or actions "should" work, causing them to skip verification steps or terminate early. This mode is typical in models with strong semantic priors.

Overall, these failure modes indicate that condensed experience, when overly abstract, generic, or mismatched, can mislead agents by reinforcing ungrounded assumptions rather than aiding decision-making. Representative cases for each failure mode are provided in Appendix D.6.

### 5.2. Suppression by Internal Biases

Beyond the semantic limitations, unfaithfulness may also arise from internal processing biases that prevent the agent from effectively using retrieved memory. To examine this possibility, we analyze how information from different prompt segments propagates through the backbone's layers and contributes to predictions (Simonyan et al., 2013).

**Probing Method.** We apply Integrated Gradients (IG) (Wang et al., 2023; Tang et al., 2025) to quantify how much each prompt segment influences the model's output. For the $h$-th head in layer $l$, we compute:

$$\text{IG}_{h,l} = A_{h,l}^T \odot \left| \frac{\partial \mathcal{L}_\theta(Y|X)}{\partial A_{h,l}} \right|, \qquad (1)$$

$$\text{IG}_{h,l}^{(r)} = \frac{1}{|\mathcal{T}_s|} \sum_{x_i \in \mathcal{T}_s} \sum_{y_j \in Y} \text{IG}_{h,l}[i,j]. \qquad (2)$$

where $\mathcal{T}_s$ denotes a specific prompt segment and $\text{IG}_{h,l}[i,j]$ measures how token $x_i$ influences token $y_j$. We then aggregate $\text{IG}_{h,l}^{(r)}$ over all heads and layers to derive a global attribution score $\text{IG}^{(r)}$, which captures the segment's overall influence on the model's prediction. A low score suggests that the model largely disregards this segment.

We perform this analysis under the ExpeL framework with Qwen3-series models, segmenting each prompt into: (1) System Instruction, (2) Condensed Experience, (3) Raw Experience, and (4) Current Trajectory.

**Results & Analysis.** Figure 7 shows results on ALFWorld using Qwen3-32B under three conditions: no intervention, Corrupt, and Irrelevant. Detailed settings and additional results on model sizes and interventions are provided in Appendix D.7. Three consistent patterns emerge:

**Condensed experience remains underutilized.** In all three settings, the IG scores attributed to condensed experience remain consistently low across layers, suggesting limited integration of such inputs regardless of their semantic quality.

**Raw experience shows stable faithful usage.** Despite the low influence of condensed content, raw experience maintains a moderate and stable contribution, further supporting our earlier findings of its reliable utilization.

**Current trajectory dominates later layers.** The most significant attribution consistently comes from the current trajectory segment, highlighting a strong local-context bias in prediction (Li et al., 2023; An et al., 2024).

In summary, these trends suggest that even when presented with semantically rich (or intervened) summaries, condensed experience struggles to influence downstream behavior due to structural biases in the model's attention flow.

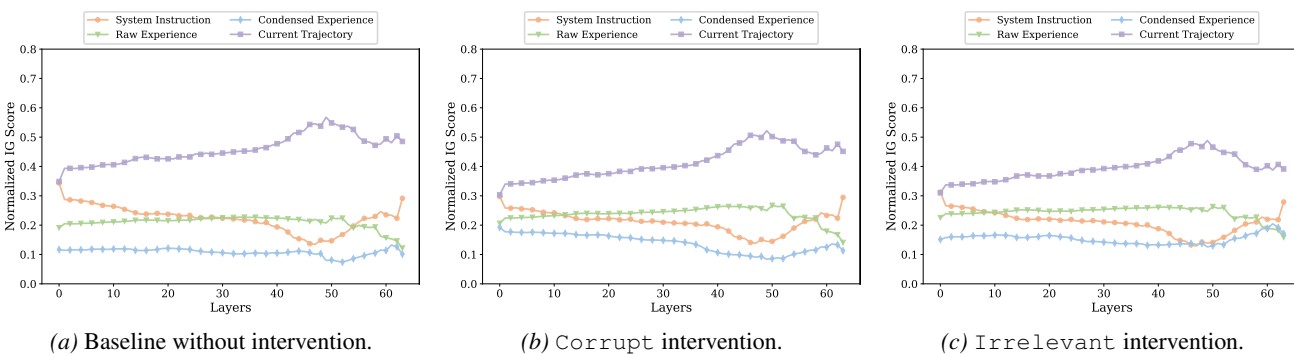

*Figure 7.* Layer-wise Integrated Gradients attribution under the ExpeL framework using Qwen3-32B. Prompts are divided into four segments: System Instruction, Condensed Experience, Raw Experience, and Current Trajectory.

### 5.3. Task-Specific Dependence on Experience

Finally, we analyze whether the task environment itself reduces the need for external experience. Many benchmarks in which agents exhibit low sensitivity to interventions, such as HotpotQA, GPQA, MMLU, and FEVER, primarily involve knowledge-intensive question answering. In these tasks, pretrained models already possess substantial factual knowledge and strong semantic priors (Shi et al., 2024), leaving retrieved experience with limited marginal value.

To further verify this, we evaluate two multi-hop question answering benchmarks: 2WikiMultiHopQA (Ho et al., 2020) and Musique (Trivedi et al., 2022). We apply interventions on ExpeL with Qwen3-32B and results are shown in Table 2. Each benchmark contributes 100 sampled examples, and we report *exact match* as the evaluation metric. More results on Qwen3-14B are in Appendix D.8.

The results demonstrate that agents still exhibit limited faithfulness to both raw and condensed experience. This suggests that the utility of experience is highly task-dependent: when a task can be handled with the agent's innate knowledge, the retrieved experience contributes little and thus fails to exert meaningful causal influence.

## 6. Related Works

**Experience-Driven Self-Evolving Agents.** Recent advances in self-evolving agents can be broadly categorized into two paradigms based on how experience is collected and utilized: *offline* and *online* approaches (Liu et al., 2025).

Offline self-evolving agents construct memory from pre-collected datasets and utilize it in a fixed form during inference (Li & Qiu, 2023; Yang et al., 2023; Zhong et al., 2024; Zhao et al., 2024a; Fu et al., 2024; Zhou et al., 2025; Yang et al., 2025b). Representative systems such ExpeL rely on pre-stored examples of successful trajectories and distilled insights without memory updates at test time.

In contrast, online paradigms allow experience memory to

*Table 2.* Intervention results on two multi-hop question answering tasks under the ExpeL framework with Qwen3-32B backbone. The evaluation metric is exact match.

| | **2Wiki-MultiHopQA** | **Musique** |
|---|---|---|
| ExpeL | 62 | 48 |
| *Raw Experience Intervention* | | |
| Empty | 65 | 43 |
| Shuffle | 63 | 46 |
| Irrelevant | 63 | 47 |
| *Condensed Experience Intervention* | | |
| Empty | 63 | 47 |
| Corrupt | 64 | 44 |
| Irrelevant | 65 | 45 |
| Filler | 64 | 45 |

evolve through interaction: agents dynamically accumulate, retrieve, and refine experiences during deployment (Chen et al., 2024; Zhang et al., 2025c; Suzgun et al., 2025). For instance, ReasoningBank (Ouyang et al., 2025) continuously distills reasoning patterns from recent episodes to enrich future responses. G-Memory (Zhang et al., 2025a) further extends this paradigm to multi-agent settings.

While these methods offer promising pathways for autonomous improvement, little attention has been paid to whether agents faithfully rely on the retrieved experiences—a gap our work aims to address.

**Faithfulness of Language Models.** Early studies on faithfulness focus on in-context learning of LMs, where the goal is to assess whether LMs genuinely leveraged human-curated in-context examples to guide their predictions (Min et al., 2022; Ye & Durrett, 2022; Shi et al., 2024). With the emergence of chain-of-thought (CoT) prompting (Wei et al., 2022), attention shift to the faithfulness of the reasoning process itself. A key line of research investigates whether the generated CoT rationale truly reflects the model's internal

decision-making, or merely serves as a post-hoc justification (Lanham et al., 2023; Turpin et al., 2023; Arcuschin et al., 2025; Lewis-Lim et al., 2025; Chen et al., 2025). In contrast, we study faithfulness in self-evolving agentic settings, where experience is both dynamic and multifaceted, revealing emerging phenomena beyond static prompt usage.

# 7. Conclusion

This work provides the first comprehensive assessment of whether self-evolving LLM agents faithfully rely on the experiences they accumulate. Through controlled interventions across raw and condensed experience, we reveal a persistent and systematic gap: agents reliably depend on raw trajectories but frequently overlook or misinterpret condensed summaries. Our deeper analysis attributes this unfaithfulness to the limited specificity of condensed content, internal processing biases of backbone, and task regimes where pretrained priors alone suffice. These findings call for future agents to incorporate experience in a way that is not only effective but also faithfully grounded.

# Acknowledgements

We thank the anonymous reviewers for their constructive comments and suggestions. This work was supported by the National Natural Science Foundation of China (NSFC) via grant 62441614 and 62576125, the Singapore Ministry of Education (MOE) Academic Research Fund (AcRF) Tier 1 grant (Proposal ID: 24-SIS-SMU-002).

# Impact Statement

This work provides the first systematic investigation into the faithfulness of experience utilization in self-evolving LLM agents. Our findings reveal a persistent and overlooked asymmetry: while agents reliably exploit raw experience, they frequently neglect or misinterpret condensed experience—even when it is the only form of guidance available. These results call into question common assumptions underlying memory-based adaptation and raise important considerations for future development of reliable and controllable agents, especially in high-stakes environments.

Our analysis offers two concrete design takeaways for future self-evolving systems. First, it highlights the importance of carefully designing the content structure of condensed experience. Rather than abstract summaries or generic advice, effective condensed experience should be contextualized, task-relevant, and cognitively actionable. This finding resonates with the concept of context engineering (Mei et al., 2025; Hua et al., 2025), which reveals that the contextual inputs play a pivotal role in shaping agent behavior. Condensed experience, as a core component of this context,

must therefore be engineered with equal care—not only to compress, but to preserve and transmit behavioral utility. This opens up promising directions for automatic condensation methods that optimize for alignment and usability, rather than surface brevity alone (Zhai et al., 2025).

Second, our findings emphasize the need to rethink the timing of experience integration. Static prepending of memory to the input—irrespective of task, timing, or agent state—can lead to underutilization or even performance degradation. Instead, experience should be retrieved and injected dynamically, based on task demands, interaction history, and internal model uncertainty. Moreover, agents may not need experience for every task; indiscriminate use can dilute attention and reduce effectiveness. Incorporating experience as a interactively-triggered signal holds promise for more faithful and efficient adaptation (Jin et al., 2025; Zhang et al., 2025d; Zhao et al., 2026).

Together, these insights lay the groundwork for a new generation of self-evolving agents that are not only capable of learning from experience, but doing so in a manner that is faithful and behaviorally grounded.

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

# A. Self-Evolving Agents

We present detailed description of the four experience-driven self-evolving agents used in our experiments: ExpeL (Zhao et al., 2024a), Dynamic Cheatsheet (Suzgun et al., 2025), ReasoningBank (Ouyang et al., 2025) and G-Memory (Zhang et al., 2025a). Across all frameworks, agents adapt their behavior by accumulating, retrieving, and reusing external experiences stored in explicit memory structures, rather than by updating the parameters of the underlying language model. In the following, we summarize the core design principles and memory mechanisms of these agents in detail:

- **ExpeL** is an offline self-evolving agent that enables a fixed language model to acquire and reuse experience across tasks without parameter updates. During an offline training phase, the agent interacts with a set of tasks and collects both successful and failed trajectories generated through trial and error. These experiences are stored in an experience pool and subsequently processed to extract natural-language insights that summarize effective strategies and general decision-making principles. In addition to abstracted insights, successful trajectories are retained to support example-based recall. At inference time, the agent augments the model's context with the extracted insights and retrieved successful experiences that are semantically similar to the current task. This design separates experience accumulation and knowledge extraction from deployment, allowing the agent to improve its task-solving behavior through offline experiential learning while keeping the underlying language model fixed during evaluation.

- **Dynamic Cheatsheet** is an online self-evolving agent that equips a fixed language model with an external memory updated during inference. The agent maintains a compact memory consisting of reusable solution elements, such as heuristics, partial solution templates, and short code fragments, derived from its own prior generations. After each task, the agent examines the produced output and selectively updates the memory by preserving entries that appear useful, while modifying or removing ineffective ones. When handling subsequent inputs, the agent retrieves relevant memory contents and incorporates them into the prompt context to influence generation. This iterative process of generation, memory update, and retrieval forms a closed-loop mechanism that enables test-time behavioral adaptation over a sequence of tasks, without performing parameter updates or altering the underlying language model.

- **ReasoningBank** embodies an instance of an online self-evolving agent. It keeps an ever-expanding memory that records condensed reasoning patterns distilled from the agent's past interactions, covering both successful cases and failures. Following each task completion, the agent assesses its own performance and selectively incorporates newly acquired experiences into this memory repository. During inference, pertinent reasoning strategies are retrieved from the memory and incorporated into the agent's context to guide future interactions. The system forms a closed feedback loop where experiences are continuously accumulated, accessed, and reused throughout deployment, enabling the agent's behavior to progressively adapt over time.

- **G-Memory** is an online self-evolving memory mechanism designed for multi-agent systems. It maintains a shared, persistent memory that records past multi-agent collaboration experiences across tasks, capturing both abstract insights and condensed interaction histories. When a new task arrives, relevant memory entries are retrieved and selectively injected into the contexts of different agents to support coordination and reasoning. After task completion, newly generated interactions and distilled insights are incorporated into the memory, updating its contents during deployment. This continual retrieval-and-update process enables agent teams to accumulate and reuse collaborative experience over time, allowing collective behavior to adapt without modifying the underlying language models.

# B. Environment and Benchmark

## B.1. Knowledge-Intensive Question Answering

- **HotpotQA** (Yang et al., 2018) is a large-scale question answering benchmark designed to support multi-hop reasoning over natural language text. It contains question–answer pairs constructed from Wikipedia articles, where answering each question requires reasoning across multiple supporting documents. The questions are diverse in form, including standard factoid queries as well as comparison and yes/no questions, and are not constrained by predefined knowledge base schemas. We selected 100 questions from the distractor dev split following ExpeL for our experiments.

- **FEVER** (Thorne et al., 2018) is a large-scale benchmark for claim verification against textual evidence. It consists of 185,445 human-written claims derived from Wikipedia, each labeled as *SUPPORTED*, *REFUTED*, or *NOT ENOUGH INFO*. The claims are generated by mutating original Wikipedia statements and are verified in a separate annotation

process, resulting in a collection that emphasizes evidence retrieval and textual entailment over natural language. Following G-Memory, we selected the first 100 questions of the validation set for our experiments.

- **GPQA-Diamond** (Rein et al., 2024) is a carefully curated and challenging subset of the Graduate-Level Google-Proof Q&A (GPQA) benchmark. It comprises 198 expert-validated questions drawn from the natural sciences, including biology, chemistry, and physics. The questions are constructed to minimize the reliance on straightforward fact recall and instead emphasize a deeper conceptual understanding. All questions can be correctly answered by domain specialists, while they are frequently challenging for non-experts, reflecting the need for complex, multi-step reasoning rather than surface-level knowledge.

- **MMLU-Pro Eng.** (Wang et al., 2024) is a professional-level subset of the MMLU benchmark that focuses on topics in physics and engineering. The questions are all formatted as multiple-choice problems. The full dataset includes 1,299 questions in physics and 969 questions in engineering, from which we randomly selected 100 questions for our experiments. The problem content spans a range of subfields within the two disciplines and requires precise technical understanding to distinguish among closely related answer options.

## B.2. Mathematical Reasoning

- **AIME 2024** The American Invitational Mathematics Examination (AIME) is a well-established benchmark derived from a prestigious high-school mathematics competition. The competition is known for its integer-answer format and time-limited setting, which further emphasizes precision and logical rigor. It consists of 133 challenging problems covering algebra, combinatorics, number theory, geometry, and probability, each requiring non-trivial mathematical reasoning and multi-step solution processes.

- **Game of 24** (Yao et al., 2023a; Suzgun & Kalai, 2024) is a heuristic-oriented arithmetic task in which the goal is to construct an expression that evaluates to 24 by using four given numbers exactly once. The task features a small combinatorial search space but allows for diverse solution paths depending on operator ordering and grouping. For example, given the input "7 7 8 11" a valid solution is "$8 \times (7 + 7 - 11)$." Solving this task requires systematic exploration of the solution space, along with strategic reasoning and pattern recognition. We adopt the 100 instances provided by (Suzgun & Kalai, 2024) to iteratively refine strategies across repeated attempts.

## B.3. Embodied Action

- **ALFWorld** (Shridhar et al., 2021) is an embodied benchmark that aligns abstract, text-based environments with interactive visual-based scenes to execute household tasks. It builds on the ALFRED (Shridhar et al., 2020) benchmark by providing paired representations of the same underlying tasks, where agents can operate through high-level textual commands in a simulated environment. The tasks span multiple categories such as pick-and-place, cleaning, heating, and cooling, and require multi-step interaction with objects and receptacles distributed across diverse room layouts. The benchmark is constructed to maintain a shared underlying world state across modalities, enabling consistent correspondence between language-level actions and embodied executions. We utilized the 134 solvable tasks.

## B.4. Web Interaction

- **WebArena** (Zhou et al., 2024) is a realistic and reproducible benchmark for web-based interaction tasks grounded in natural language instructions. WebArena provides 812 web navigation tasks that cover four common application domains: Shopping (187), CMS (182), Reddit (106) and Map (109). BrowserGym (de Chezelles et al., 2025) is used as the execution environment for WebArena. Each task is specified as a high-level natural language intent and requires agents to execute a sequence of concrete web interactions across dynamic pages and tools. The benchmark includes a curated set of long-horizon tasks with programmatic validation based on functional correctness of the final outcomes.

- **WebShop** (Yao et al., 2022) is a large-scale web interaction benchmark that simulates realistic online shopping scenarios through a self-contained e-commerce environment. It includes over one million real-world products and 12,087 crowdsourced natural language instructions, each specifying a product requirement to be fulfilled through a sequence of web-based actions. Given an instruction, an agent must navigate search results, inspect product pages, select appropriate options, and complete a purchase to satisfy the specified constraints. We follow the instructions of ExpeL to set the implementation details of the WebShop Environment. For WebShop tasks, we evaluated using the same 100 tasks used by ReAct (Yao et al., 2023b), Reflexion (Shinn et al., 2023) and ExpeL.

# C. Implementation Details

- **ExpeL** We follow the official setup described in the ExpeL (Zhao et al., 2024a). For agent hyperparameters, the LLM decoding temperature is set to 0.0, and greedy decoding is used. During the experience gathering stage, the maximum number of reflection retries is set to 3. At evaluation time, ExpeL retrieves the top-$k$ most similar successful trajectories and uses them as in-context demonstrations. The vector store is implemented with Faiss, the retriever uses kNN, and all task descriptions are embedded using the *all-mpnet-base-v2* encoder.

  Different benchmarks adopt different interaction budgets and retrieval strategies. Specifically, for HotpotQA, each task is allowed up to 7 environment interaction steps, and up to 6 successful trajectories are retrieved as few-shot demonstrations during evaluation. For WebShop, each task allows up to 15 interaction steps, with up to 2 retrieved successful trajectories. For ALFWorld, each task allows up to 20 steps, and up to 2 successful trajectories are retrieved.

- **Dynamic Cheatsheet** We follow the official experimental setup of Dynamic Cheatsheet (Suzgun et al., 2025). For agent hyperparameters, the LLM decoding temperature is set to 0.0, and greedy decoding is used. Across all benchmarks in Dynamic Cheatsheet, including AIME 2024, Game of 24, GPQA-Diamond, and MMLU-Pro Eng., we adopt the same retrieval configuration. Specifically, before each new task, the system computes cosine similarity between the embedding of the current query and embeddings of historical queries, and retrieves the top-3 most similar past input–output pairs from memory to support inference.

- **ReasoningBank** Our experimental setup largely follows the configuration of ReasoningBank (Ouyang et al., 2025). For the agent, we set the decoding temperature to 0.7 and adopt greedy decoding as the decoding strategy. On the WebArena benchmark, each task is allowed a maximum of 30 interaction steps. We use *text-embedding-ada-002* as the embedding model to encode queries and memory items, and employ cosine similarity for retrieval. For each new query, the agent retrieves the top-3 most relevant memory items, which are then injected into the agent's prompt.

  Task outcomes are labeled as success or failure using LLM-as-a-judge (Zheng et al., 2023). The judge model, as well as the model used for memory extraction, is configured with a decoding temperature of 0.0, with the goal of maximizing determinism in the evaluation and extraction process. Each memory item follows a fixed schema consisting of three components: a Title, which concisely summarizes the underlying strategy; a Description, which provides a one-sentence abstract of the memory; and Content, which contains one to five sentences of distilled insights.

- **G-Memory** We follow the experimental setup of G-Memory (Zhang et al., 2025a). For coarse-grained retrieval over the query graph, queries are encoded using a MiniLM sentence embedding model and matched with cosine similarity, retrieving the top-$k$ most similar historical queries. For fine-grained retrieval, we further select the top-$M$ relevant queries using an LLM-based relevance scorer and sparsify their interaction graphs with an LLM-based graph compression module. The values of $k$ and $M$ are treated as tunable hyperparameters. Query nodes are labeled with execution status from Failed, Resolved. All retrieved insights and interaction subgraphs are injected into agent prompts before task execution, and the memory graphs are updated after each task without any gradient-based training.

# D. Details of Intervention Design

We provide additional details and design motivations for the intervention strategies used in our faithfulness evaluation. As discussed in §3, raw and condensed experiences differ substantially in their structure, abstraction level, and expected utility. Our intervention design reflects these differences while adhering to two principles: (1) targeting the failure modes most likely to occur in each experience type, and (2) ensuring minimal disruption to input formatting and inference pipelines. To concretize these designs, Table 3 presents illustrative examples of each intervention applied to ExpeL framework.

## D.1. Raw Experience Interventions

Raw experiences consist of complete trajectories, often encoded as multi-step sequences of observations, actions, and outcomes. They are typically grounded in specific tasks and retain temporal and causal coherence.

- **Empty:** This intervention removes the semantic content of the raw experience (e.g., deleting trajectory steps) while retaining the prompt structure (e.g., "Here are two examples of past successful trajectories:"). This allows us to test whether the agent benefits from the actual content or merely from the presence of a scaffolded context block.

*Table 3.* An example of experience interventions.

| **System Instruction** | |
|---|---|
| **Task Goal** | Question: Why did Grand Duke Kirill Vladimirovich Of Russia's wife die? |

| **Raw Experience Interventions** | |
|---|---|
| **Original** | Here are some examples:
Question: Who died first, Fleetwood Sheppard or George William Whitaker?
Action 1: Search[Fleetwood Sheppard]
Observation 1: Fleetwood Sheppard (1 January 1634 – 25 August 1698) was an English courtier.
Action 2: Search[George William Whitaker]
Observation 2: George William Whitaker (September 25, 1840 – March 6, 1916) was a painter.
Action 3: Finish[Fleetwood Sheppard]
Observation 3: Answer is CORRECT |
| **Empty** | Here are some examples: |
| **Shuffle** | Here are some examples:
Question: Who died first, Fleetwood Sheppard or George William Whitaker?
Observation 3: Answer is CORRECT
Action 3: Finish[Fleetwood Sheppard]
Action 1: Search[Fleetwood Sheppard]
Observation 1: Fleetwood Sheppard (1 January 1634 – 25 August 1698) was an English courtier.
Action 2: Search[George William Whitaker]
Observation 2: George William Whitaker (September 25, 1840 – March 6, 1916) was a painter. |
| **Irrelevant** | Here are some examples:
Question: What is the place of birth of Clara Novello's father?
Action 1: Search[Pavel Urysohn]
Observation 1: Pavel Samuilovich Urysohn was a Soviet mathematician.
Action 2: Search[Leonid Levin]
Observation 2: Leonid Anatolievich Levin is a mathematician and computer scientist.
Action 3: Finish[yes] |

| **Condensed Experience Interventions** | |
|---|---|
| **Original** | The following are some experience you gather on a similar task. Use these as references to help you perform this task:
1. When encountering conflicting or ambiguous search results, perform a lookup for the exact name or title to verify identity and avoid confusion with similarly named individuals.
2. If a search repeatedly returns incorrect or unrelated information, systematically refine the search query by incorporating contextual qualifiers, roles, or relationships to isolate the correct individual or entity. |
| **Empty** | The following are some experience you gather on a similar task. Use these as references to help you perform this task: |
| **Corrupt** | The following are some experience you gather on a similar task of question answering using Wikipedia API. Use these as references to help you perform this task:
1. When encountering conflicting [CORRUPTED_561] ambiguous search results, perform a lookup for the exact name or [CORRUPTED_842] to verify identity and avoid confusion with similarly named individuals. [ERROR_INFO]
2. If a search repeatedly returns [CORRUPTED_746] or unrelated information, systematically refine the search query by incorporating contextual qualifiers, roles, [ERROR_INFO] or relationships isolate the individual or entity. |
| **Irrelevant** | The following are some experience you gather on a similar task. Use these as references to help you perform this task:
1. Literature contains various genres and styles.
2. Art expresses human creativity and emotion. |
| **Filler** | The following are some experience you gather on a similar task. Use these as references to help you perform this task:
1. ... $$$ ###
2. *** ... *** |

- **Shuffle:** By randomly permuting the steps in each trajectory, we disrupt its internal temporal and causal structure while preserving token-level content. This tests whether the agent relies on coherent sequencing in trajectory usage.

- **Irrelevant:** Retrieved trajectories are replaced with examples drawn from unrelated tasks with similar surface format. This targets the agent's sensitivity to topical relevance and allows us to evaluate semantic grounding.

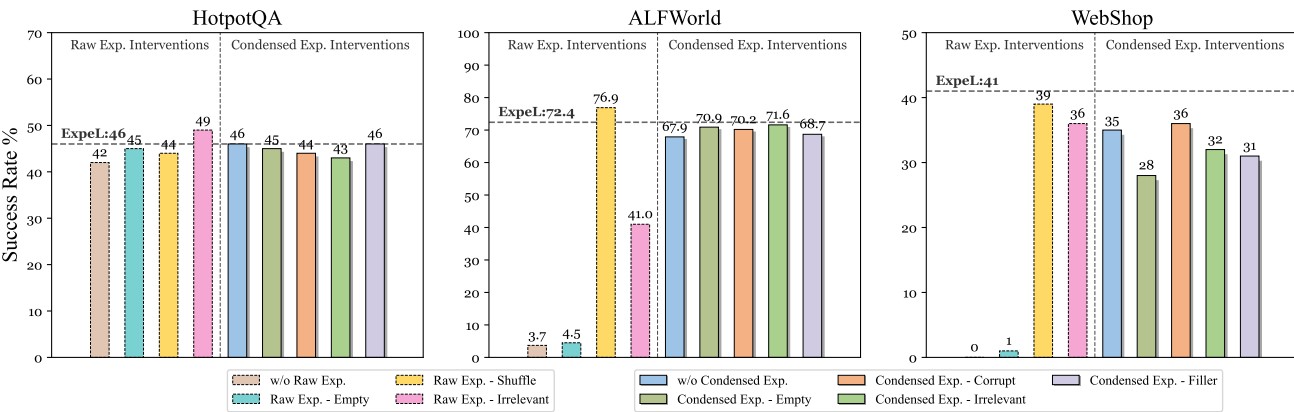

*Figure 8.* Results on Qwen3-235B-A22B for HotpotQA, ALFWorld, and WebShop under ExpeL. We observe similar patterns as in GPT-4o: strong reliance on raw experience and inconsistent faithfulness to condensed summaries.

We use three interventions for raw experience, as these cover the main axes of disruption: existence, ordering, and relevance. We avoid over-complicating the raw setting since its structure is already explicit and well-formed.

## D.2. Condensed Experience Interventions

Condensed experiences are distilled textual summaries derived from past interactions, often presented as high-level heuristics, plans, or failure abstractions. Compared to raw trajectories, these summaries are more abstract, loosely structured, and typically lack strict ordering constraints.

- **Empty:** We remove the content of the summary while preserving formatting (e.g., "Here is a distilled insight:"), in order to test whether performance is attributable to semantic content or merely to the presence of a template.

- **Corrupt:** Key logical elements are randomly perturbed (e.g., inverting causal relations, altering conditionals, or replacing verbs), breaking the internal coherence while maintaining surface form. This helps assess whether the agent truly parses and applies the intended reasoning patterns.

- **Irrelevant:** A summary from an unrelated task is inserted in place of the retrieved one, preserving general topic style while disrupting task alignment. This intervention probes reliance on contextual specificity.

- **Filler:** The entire content is replaced with semantically meaningless placeholder tokens (e.g., sequences of special characters such as "%$#@&"). This isolates whether improvements stem from semantic value.

Unlike raw trajectories, condensed summaries do not rely on internal ordering of steps or temporal structure. Therefore, we do not include a `Shuffle` variant in this setting. Instead, we introduce two complementary perturbations:

`Corrupt` targets internal *semantic consistency*, testing whether the agent interprets specific logic embedded in the summary. `Filler` targets *semantic emptiness* under preserved surface form, to assess reliance on textual formatting. These design choices reflect the unique failure modes of abstracted experience and enable more fine-grained diagnosis of when and how condensed content influences behavior.

## D.3. Additional Results on Qwen3-235B-A22B and Claude-Sonnet-4.6 with ExpeL

To verify the consistency of our findings across backbone models, we replicate the same intervention study using the open-weight Qwen3-235B-A22B and the frontier closed-source model Claude-Sonnet-4.6. The results, visualized in Figure 8 and Figure 9, align closely with the GPT-4o-based findings.

First, agents remain highly sensitive to raw experience interventions. Across HotpotQA, ALFWorld, and WebShop, the removal of raw experience (especially `Empty` and `Irrelevant`) consistently results in substantial performance drops, reaffirming that the agent does not simply rely on priors but indeed benefits from retrieved raw trajectories. This trend

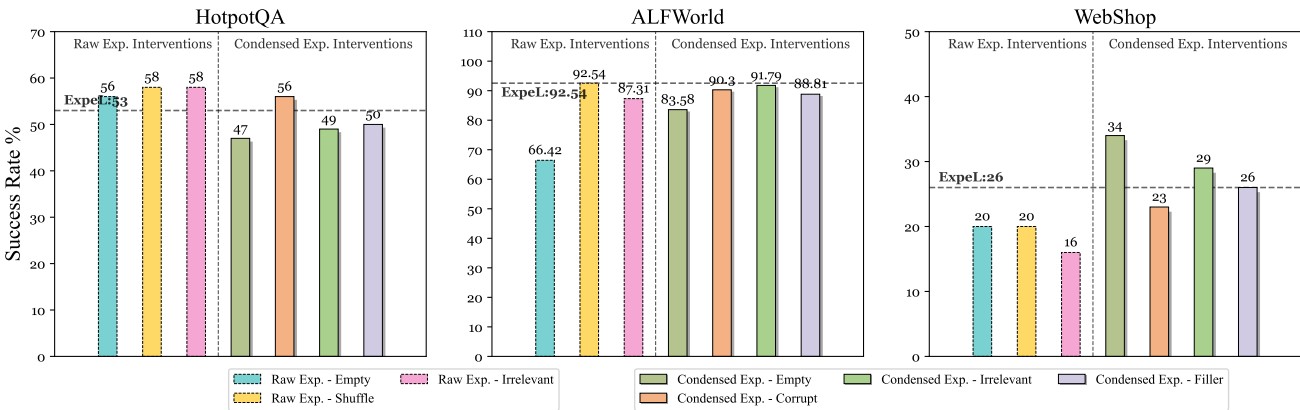

*Figure 9.* Results on Claude-Sonnet-4.6 for HotpotQA, ALFWorld, and WebShop under ExpeL. We observe similar patterns as in GPT-4o: strong reliance on raw experience and inconsistent faithfulness to condensed summaries.

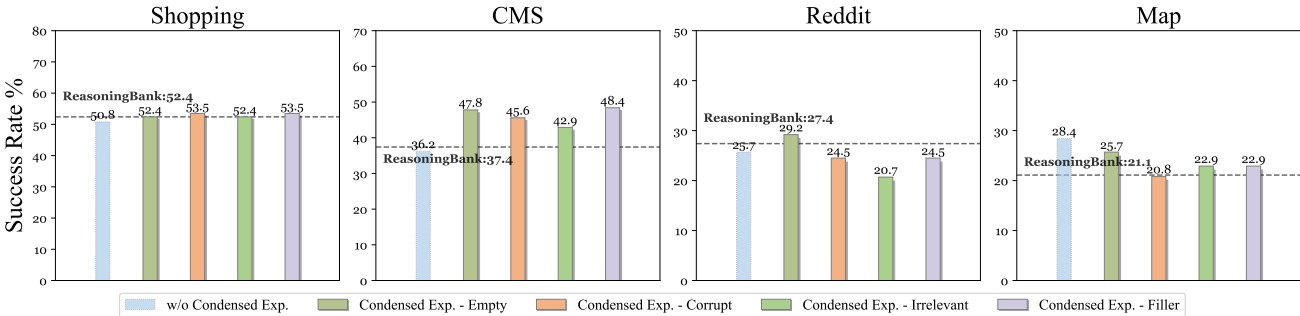

*Figure 10.* Condensed-only performance on the ReasoningBank framework (online, single-agent) using Qwen3-14B. Perturbing condensed experience has limited or inconsistent impact across tasks.

remains evident even for stronger frontier models such as Claude-Sonnet-4.6, suggesting that scaling model capability does not fundamentally resolve the reliance on raw experience.

Second, the faithfulness to condensed experience remains weak. In both Qwen3-235B-A22B and Claude-Sonnet-4.6, perturbing condensed summaries often yields only minor or inconsistent performance changes. For example, on HotpotQA under Claude-Sonnet-4.6, all condensed interventions remain close to the original ExpeL performance (53→47–56), while raw interventions exhibit much larger fluctuations. Similarly, on ALFWorld and WebShop, condensed perturbations only induce limited degradation compared with raw experience interventions. These results further strengthen our main conclusion that current self-evolving agents remain substantially less faithful to condensed experience than to raw trajectories, even with stronger backbone models.

Lastly, we observe that `Shuffle` (raw) and `Filler` (condensed) preserve surface structure while destroying coherence or semantics. The agent's partial robustness to these reveals possible overreliance on format cues rather than deep content integration, further motivating the need for more faithful mechanisms of experience exploitation.

### D.4. Additional Results on Condensed-Only Setting: ReasoningBank

**Overview.** In the main text, we demonstrate that agents exhibit limited faithfulness to condensed experience when both raw and condensed memories are present. However, one may wonder: is this apparent unfaithfulness due to reliance on raw experience, overshadowing the role of condensed summaries? To answer this, we evaluate agents in a **condensed-only** setting using the ReasoningBank, which provides only distilled insights from past trajectories without access to raw examples.

**Qwen3-14B and Qwen3-32B exhibit consistent trends.** Figure 10 and Figure 11 shows results of Qwen3-14B and Qwen3-32B on four WebArena tasks under the ReasoningBank setup, respectively. Despite architectural and scale differences, both models display similar behavior to the Gemini-2.5-Flash (see main Figure 4): interventions such as `Corrupt`, `Irrelevant`, and `Filler` yield only minor performance degradation. In some cases, like Shopping and

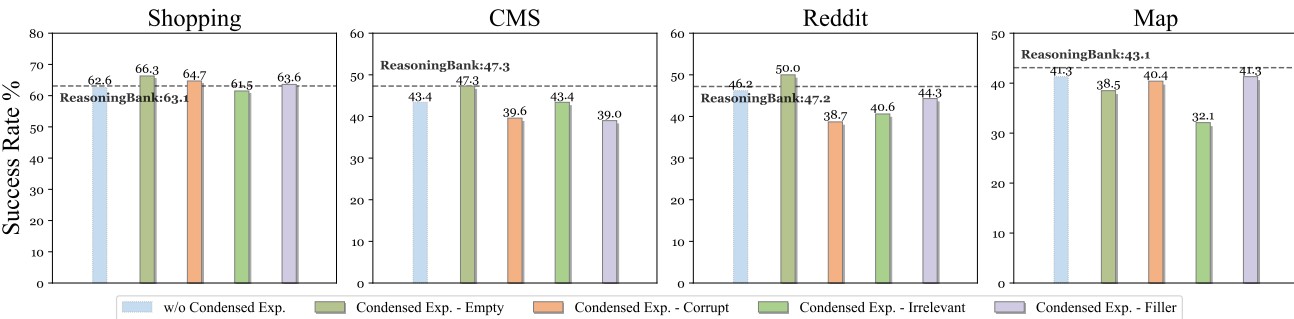

*Figure 11.* Condensed-only performance on the ReasoningBank framework (online, single-agent) using Qwen3-32B. Similar to the 14B variant, condensed interventions do not strongly influence success rates.

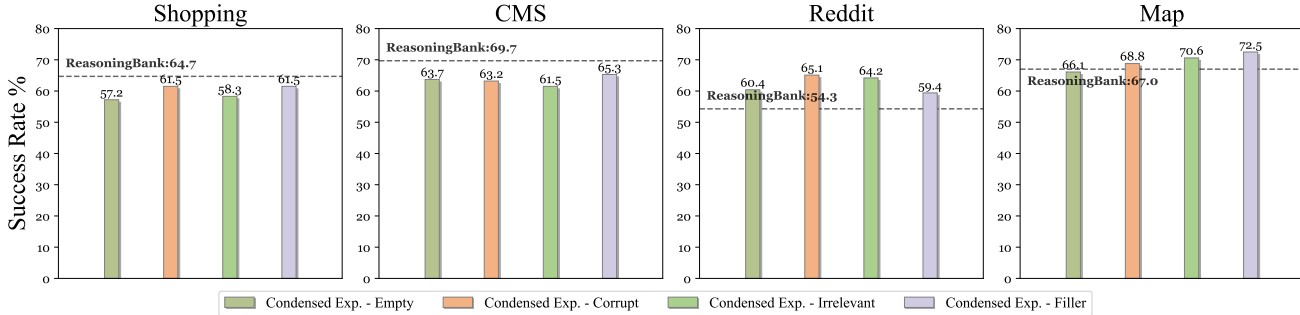

*Figure 12.* Condensed-only performance on the ReasoningBank framework (online, single-agent) using Gemini-3-Pro. Similar to the 14B variant, condensed interventions do not strongly influence success rates.

CMS, success rates remain comparable or even slightly improve under perturbed conditions.

**Gemini-3-Pro shows the same pattern.** To further validate the generality of this phenomenon on stronger frontier models, we additionally evaluate Gemini-3-Pro under the same condensed-only ReasoningBank setup. As shown in Figure 12, the model exhibits behavior highly consistent with both Qwen3 variants and Gemini-2.5-Flash. Across Shopping, CMS, Reddit, and Map, interventions such as `Corrupt`, `Irrelevant`, and `Filler` induce only limited or inconsistent performance changes, with several perturbed settings even slightly outperforming the original configuration. These results further strengthen our conclusion that current agents often fail to faithfully utilize the semantic content of condensed experience, even when such experience is the only available external guidance.

These findings reinforce our earlier conclusion: even when condensed experience is the sole retrieved knowledge, agents often do not meaningfully exploit its semantic content. The lack of sensitivity across various perturbations—including semantically incoherent or content-free fillers—suggests superficial reliance or formatting over genuine understanding.

### D.5. Additional Results on Multi-agent Setting with Qwen3-235B-A22B and GPT-5.2

To verify the generality of our findings in multi-agent online self-evolving settings, we further replicate the G-Memory experiments using a stronger backbone: Qwen3-235B-A22B. The results on FEVER and ALFWorld are shown in Figure 13.

**Faithfulness to raw experience holds across sources and models.** In the ALFWorld domain, removing or corrupting either source of raw experience (Reference or Execution) results in large performance degradation. For example, removing execution experience (`Exe-Raw Exp. Empty`) drops performance by $-6.8$ points, and replacing human-written reference experience with irrelevant ones leads to an even greater drop of $-30.6$ (from 78.4 to 47.8). This mirrors our earlier observations under GPT-4o-mini (Figure 5) and reinforces that regardless of origin, raw experiences are faithfully grounded.

Interestingly, this effect generalizes well even in a stronger model: Qwen3-235B-A22B not only follows the same degradation trend, but also amplifies the raw-condensed disparity more clearly, suggesting that as models become more capable, their selective grounding behavior becomes more pronounced.

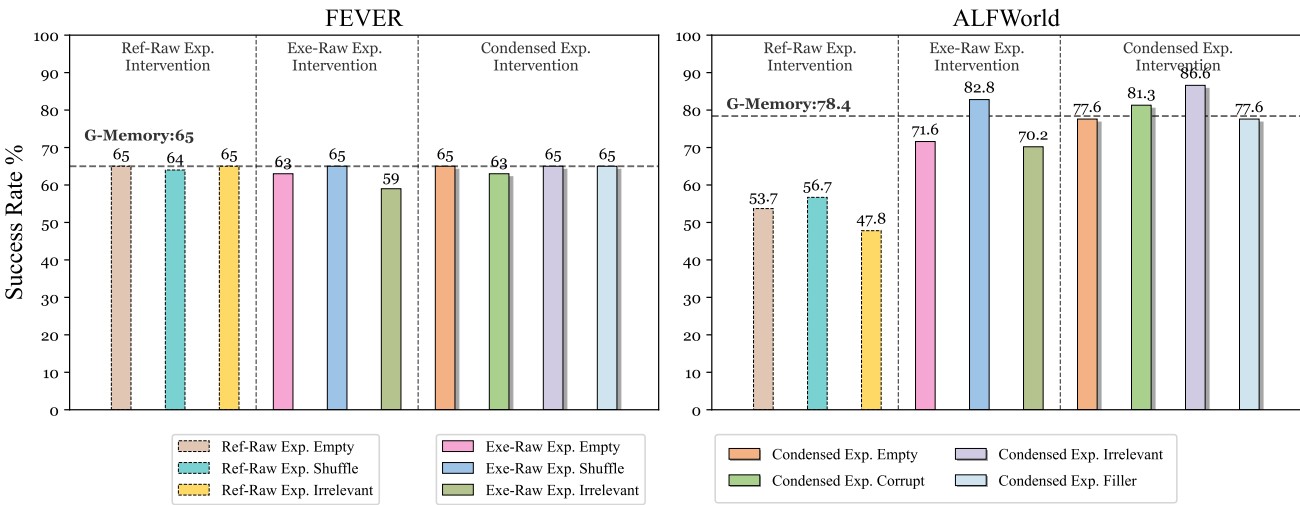

*Figure 13.* Intervention results for the G-Memory framework using Qwen3-235B-A22B as the backbone. Across both FEVER and ALFWorld, interventions on both Reference and Execution Raw Experiences cause clear performance degradation, while condensed experience perturbations have notably weaker effects. This supports the generality of our earlier conclusions.

**Condensed experience remains less semantically grounded.** Across both FEVER and ALFWorld, interventions such as `Corrupt`, `Irrelevant`, or `Filler` show limited or even no degradation (e.g., `Filler` still achieves 77.6, almost matching unperturbed performance). This further validates our earlier claim: while condensed experience offers utility, agent decisions do not strongly rely on its internal semantics, even in large-scale models.

**GPT-5.2 exhibits highly consistent behavior.** We further evaluate the G-Memory framework using the stronger frontier backbone GPT-5.2. As shown in Figure 14, the same overall pattern continues to hold across both FEVER and ALFWorld. Interventions on raw experience—whether Reference or Execution trajectories—still lead to clear performance degradation. For example, on ALFWorld, perturbing execution raw experience reduces performance from 75.4 to 47.0 (`Exe-Raw Exp. Shuffle`), while modifying reference raw experience also causes substantial drops.

In contrast, interventions on condensed experience remain much less influential. Across both domains, `Corrupt`, `Irrelevant`, and `Filler` produce only limited or inconsistent changes, with several settings even outperforming the original configuration. These results further strengthen our conclusion that the raw–condensed faithfulness asymmetry persists even in stronger frontier models and collaborative multi-agent settings.

### D.6. Representative Failure Cases for Condensed Experience

To distinguish failure modes where an agent succeeds without condensed experience but fails when it is added, we group them into three categories as described in §5.1: Distraction from the Task Goal, Overreliance on Incorrect Priors, and Premature Inference from Prior Patterns. Representative cases are shown in Table 4, Table 5, and Table 6, respectively.

### D.7. Additional IG Attribution Results

**Setup.** We follow the official experimental setup of Integrated Gradients (IG) (Wang et al., 2023; Tang et al., 2025). The IG score is computed on the *last-turn prompt*, i.e., the input context immediately before the agent executes its final action. Due to computational and memory constraints in long-context settings, we adopt an approximate IG formulation for efficiency. Specifically, we use the gradient magnitude of token embeddings as a surrogate for the attention-level IG score. Prior work has shown that embedding gradient norms exhibit a strong positive correlation with attention-based IG scores across different token categories, validating this approximation (Tang et al., 2025). Concretely, for a given sample, we compute the gradients of the token embeddings with respect to the cross-entropy loss, take the L2 norm for each token embedding, and use the arithmetic mean of the L2 norms over all tokens as the final IG score for the sample.

**Results & Analysis.** To validate the generality of our findings, we extend our attribution analysis to a broader range of model sizes within the Qwen3 series. Specifically, we report layer-wise Integrated Gradients (IG) scores under the ExpeL

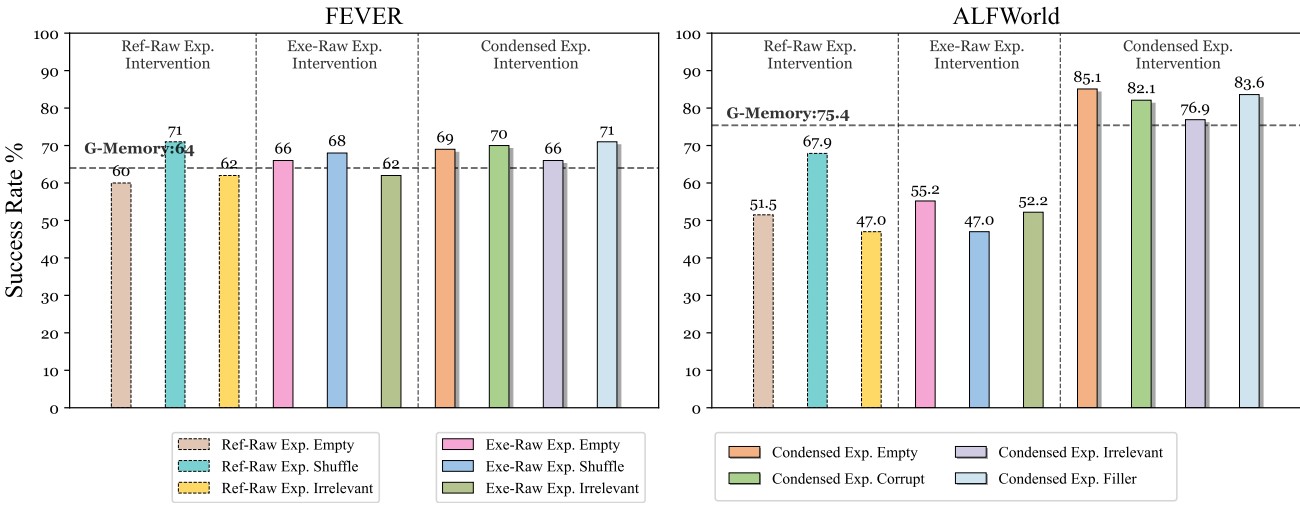

*Figure 14.* Intervention results for the G-Memory framework using GPT-5.2 as the backbone. Across both FEVER and ALFWorld, interventions on both Reference and Execution Raw Experiences cause clear performance degradation, while condensed experience perturbations have notably weaker effects. This supports the generality of our earlier conclusions.

*Table 4.* Failure case of distraction from the task goal.

| |
|---|
| **Task Goal**: How much refund I should expect from my order canlled in 2022, including shipping fee. |
| **With condensed experience**: |
| **Think**: From the memory items, I remember that accessing the order history via "My Account" is the direct way to find order details.
**Action**: click('176')
**Think**: The user wants to include shipping fees, so if the total is already the order total (which presumably includes shipping), then that's the refund amount. But to be sure, maybe we should click "View Order" to check the details. **However, Memory Item 3 advises against clicking into details if the total is visible.** Therefore, the next action is to send the user the refund amount based on the visible total.
**Action**: send_msg_to_user("The refund amount for your canceled order (Order #000000177 from 10182022) is $2,126.32, which includes shipping fees.")
**Evaluation**: **FAILED**. The bot successfully navigated to the correct order and calculated the refund amount. However, the final response to the user was incomplete, missing the full amount information. |
| **Without condensed experience**: |
| **Think**: The next step is to click on "My Account" to access their account details.
**Action**: click('176')
**Think**: The order in question is listed as canceled, and the total is $365.42. But the user wants to know the refund amount including shipping. To find out, I should click on "View Order" for that specific canceled order.
**Action**: click('584')
**Think**: The order details show the subtotal as $350.42, shipping and handling at $15.00, making the grand total $365.42.
**Action**: send_msg_to_user("The total amount for your canceled order #000000170 is $365.42, which includes a subtotal of $350.42 for the items and $15.00 for shipping.")
**Evaluation**: **SUCCEED**. The agent correctly identified the total refund amount, including the item and shipping costs, as $365.42, which matches the information provided on the webpage. The response also advises the user to check with customer support for any potential deductions, which is a reasonable caution. |

framework for Qwen3-1.7B, Qwen3-4B, Qwen3-8B, Qwen3-14B, and Qwen3-32B, across multiple intervention conditions on condensed experience. The results for these models are shown in Figure 15, Figure 16, Figure 17, Figure 18 and Figure 19, respectively. Across all scales, we observe consistent patterns:

**Condensed experience exhibits persistently low attribution across layers.** Across all Qwen3 models and settings, the IG scores assigned to condensed experience remain consistently low and flat, indicating that these segments contribute minimally to the final output. Even when the condensed experience is intervened, the model rarely shifts its attention or

*Table 5.* Failure case of overreliance on incorrect priors.

| |
|---|
| **Task Goal**: Provide me with the full names of chargers from Anker, and also share the price range for the available models. |

| |
|---|
| **With condensed experience**: |

**Think**: First, I need to search for Anker chargers. Memory Item 1 says to use the search function with precise keywords.
**Action**: fill('211', 'Anker chargers')
**Think**: The current page shows a search result for "Anker chargers," but looking at the product list, none of the items are actually Anker chargers. **Since the search results are not relevant, the next step is to handle the case where search results are incomplete or absent, as per memory item 3**. Therefore, the assistant should inform the user that no Anker chargers were found in the search results and suggest alternative stepss.
**Action**: send_msg_to_user("It seems there are no Anker charger products in the current search results. Let me check if there are other categories or refine the search further.")
**Evaluation**: **FAILED**. The agent failed to find the Anker chargers as per user's intent. The webpage shows unrelated products, indicating the search did not succeed. Moreover, the agent did not provide the requested information on the full names or price range of the chargers.

| |
|---|
| **Without condensed experience**: |

**Think**: Let me try filling that with "Anker chargers" and then see if the search button becomes enabled. But since the button is disabled, perhaps the search is triggered by pressing Enter or another action. Let me try filling the search box and then pressing Enter.
**Action**: fill('214', 'Anker chargers'); press('214', 'Enter')
**Think**: The current page shows search results for "Anker chargers," and there are several products listed. Each product has a name and a price.
**Action**: send_msg_to_user("Here are some Anker charger models and their prices from the search results: (Detailed search results). The price range for available models is from \$8.99 to \$59.99.")
**Evaluation**: **SUCCEED**. The agent successfully provided a list of Anker charger models along with their prices, matching the user's request. The price range was also accurately given.

reliance toward these inputs, suggesting a structural underutilization regardless of quality.

**Raw experience is moderately and faithfully leveraged.** In contrast, raw experience segments show stable and nontrivial attribution across layers, especially in the mid-to-deep transformer blocks. This pattern holds even under different intervention conditions and model scales, confirming that raw trajectories are causally linked to agent behavior and thus more faithfully integrated into decision-making.

**Local context dominates in deeper layers.** The IG scores for the current trajectory tokens rise sharply in the later layers, becoming the dominant source of influence. This demonstrates the model's tendency to increasingly attend to and rely on its ongoing generation context, rather than external retrieved signals, as output tokens are being produced.

**Empty intervention triggers compensatory over-attribution to the placeholder segment.** When the condensed experience is replaced with an empty string, its attribution unexpectedly increases, especially in shallow to mid layers. This suggests that the model initially treats the placeholder segment as potentially informative due to its positional slot or formatting. However, this early attribution is short-lived—quickly diminishing in deeper layers as the model fails to extract meaningful content. This transient spike, followed by sharp decay, underscores the model's structural bias to initially attend to all segments but ultimately rely on meaningful local context.

**Condensed experience receives slightly elevated attribution when perturbed, but without meaningful impact.** In some models, introducing intervention (especially irrelevant content) into the condensed experience causes small bumps in early-layer IG scores. However, these signals quickly dissipate in deeper layers, and the behavioral outputs remain largely unchanged. This suggests that while the underlying backbone model may transiently register the presence of modified input, it does not integrate it in a behaviorally significant way.

These findings reinforce the presence of position-sensitive processing biases and suggest that increasing model capacity alone is insufficient to ensure faithful usage of retrieved condensed experience.

*Table 6.* Failure case of premature inference from prior patterns.

| |
|---|
| **Task Goal**: Show the most recent processing order. |

| |
|---|
| **With condensed experience**: |

**Think**: Okay, the user wants to see the most recent processing order. The 'My Account' link in the banner is a list item with a link.
**Action**: click('175')
**Think**: The first order listed is Order #000000170 from 5/17/23 with a status of "Canceled". However, the user might want the most recent one that's processed, maybe not canceled. Wait, the user just said "most recent processing order", so maybe the latest one regardless of status. **The status is "Canceled". But perhaps the user wants to see that**.
**Action**: send_msg_to_user("The most recent processing order is Order #000000170, placed on May 17, 2023. Status: Canceled. Items: 1. Uttermost Volterra Lamp ($250.80), 2. NOZE Coat Rack ($40.99), 3. Plus Size Lingerie ($58.63). Total: $365.42. Shipping and billing address: Emma Lopez, 101 S San Mateo Dr, San Mateo, CA 94010. Payment method: Check/Money order.")
**Evaluation**: **FAILED**. The user's intent was to see the most recent processing order. The bot provided information about Order #000000170, but the status is canceled. The information on the webpage confirms this order is canceled, matching the bot's response. Therefore, the task is incomplete as it didn't provide an active processing order.

| |
|---|
| **Without condensed experience**: |

**Think**: Okay, let's see. The user wants to show the most recent processing order. The navigation menu has "My Account" as a link.
**Action**: click('177')
**Think**: The most recent order based on the date would be the one from 5/17/23, which is order #000000170. **But the user might be looking for the latest one that's not canceled**. The next one is 5/2/23 with order #000000189, status Pending.
**Action**: send_msg_to_user("The most recent processing order is Order #000000189. It is currently pending, placed on May 2, 2023. The order includes a Quoizel Tiffany Floor Lamp with a total of $754.99. The shipping and billing addresses are both for Emma Lopez in San Mateo, California. The payment method selected is Check / Money order.")
**Evaluation**: **SUCCEED**. The agent successfully retrieved and conveyed the details of the most recent processing order to the user. The information provided matches the details shown on the final webpage snapshot, ensuring accuracy and completeness of the response.

### D.8. Additional Task-Specific Dependence Results

To ensure the robustness of this trend across model scales, we further report results on Qwen3-14B (Table 7). These findings mirror the observations with Qwen3-32B. Specifically, interventions on raw experience do not significantly hurt performance—sometimes even yielding higher scores. Similarly, condensed experience interventions exhibit no consistent degradation, and in Musique, several perturbed conditions (e.g., `Corrupt` or `Filler`) even outperform ExpeL.

These patterns reinforce the conclusion that neither raw nor condensed experience is causally utilized in these multi-hop tasks—even when reasoning is involved. Instead, models exhibit strong reliance on internal priors, rendering external experience unnecessary or even misleading.

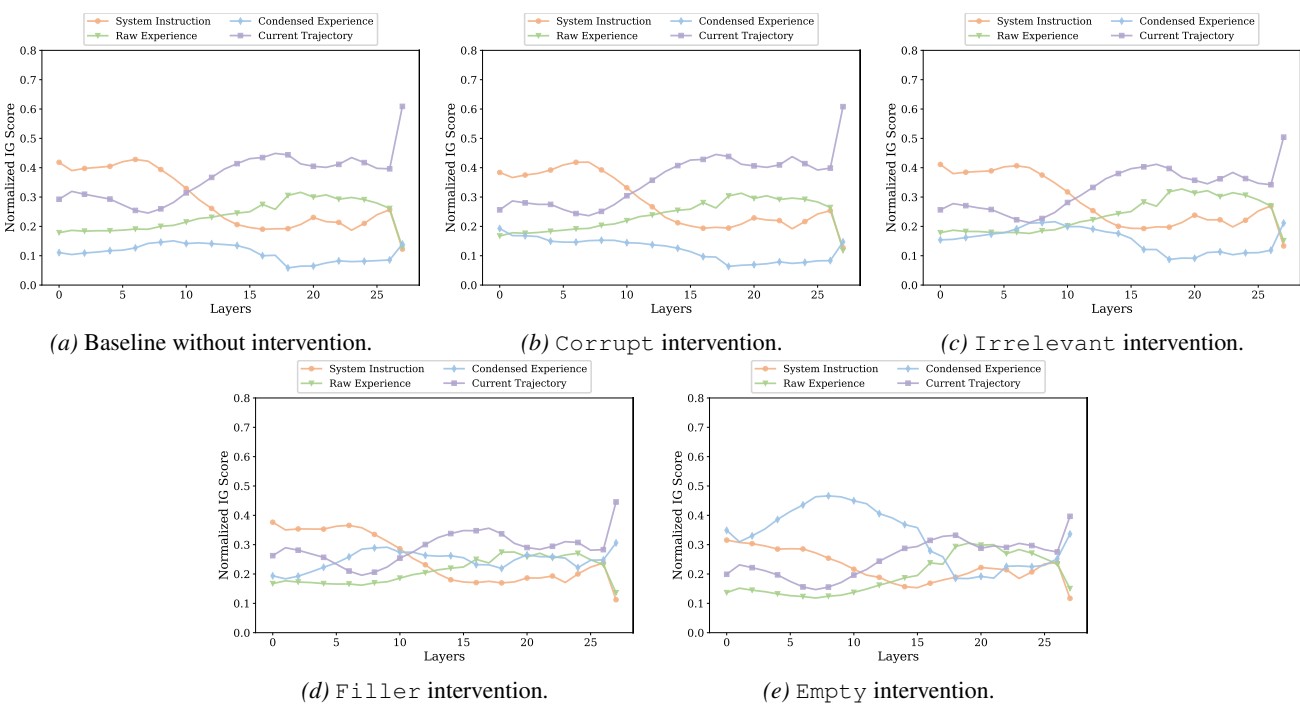

*(a)* Baseline without intervention.   *(b)* `Corrupt` intervention.   *(c)* `Irrelevant` intervention.

*(d)* `Filler` intervention.   *(e)* `Empty` intervention.

*Figure 15.* Layer-wise Integrated Gradients attribution under the ExpeL framework using Qwen3-1.7B. Prompts are divided into four segments: System Instruction, Condensed Experience, Raw Experience, and Current Trajectory.

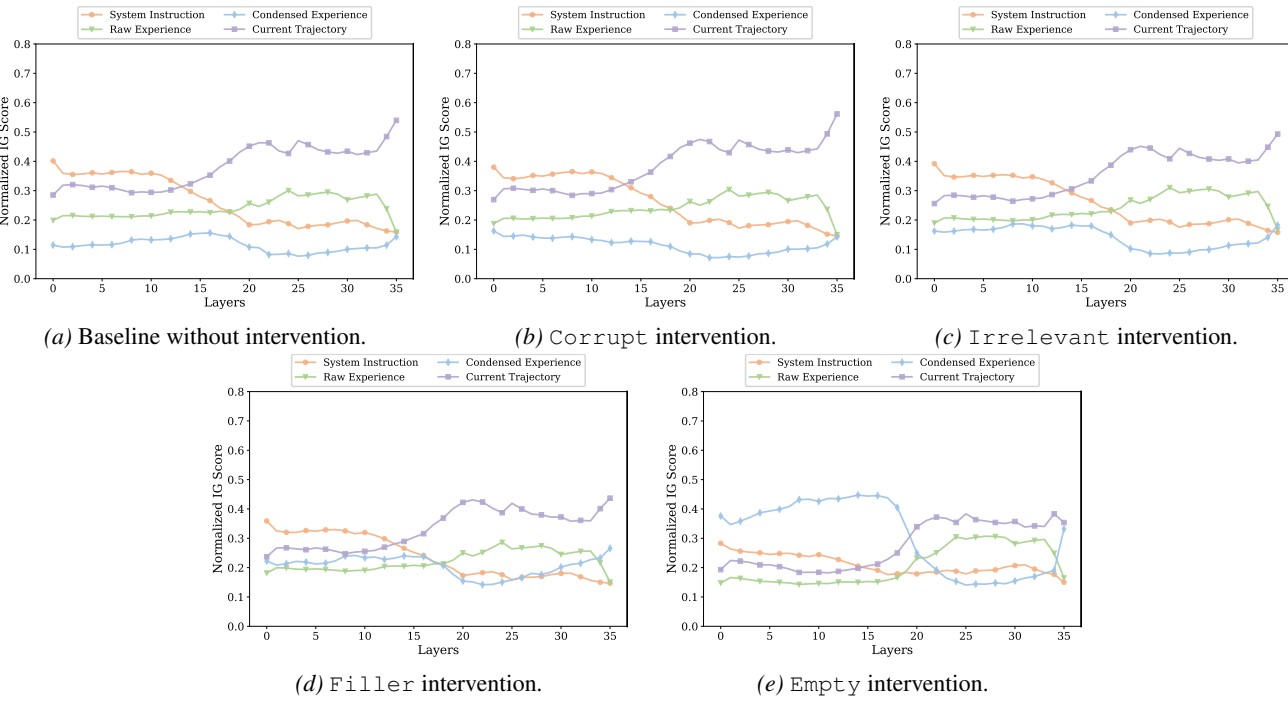

*(a)* Baseline without intervention.   *(b)* `Corrupt` intervention.   *(c)* `Irrelevant` intervention.

*(d)* `Filler` intervention.   *(e)* `Empty` intervention.

*Figure 16.* Layer-wise Integrated Gradients attribution under the ExpeL framework using Qwen3-4B. Prompts are divided into four segments: System Instruction, Condensed Experience, Raw Experience, and Current Trajectory.

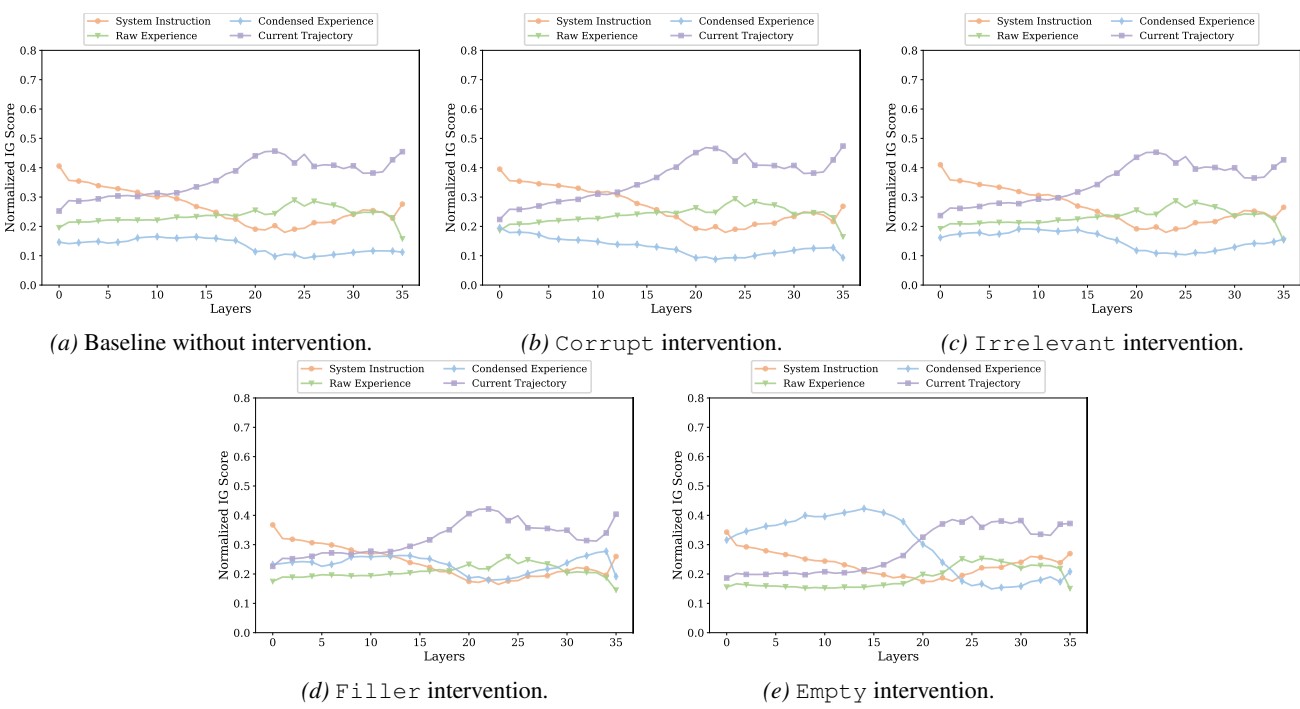

*(a)* Baseline without intervention.     *(b)* `Corrupt` intervention.     *(c)* `Irrelevant` intervention.

*(d)* `Filler` intervention.     *(e)* `Empty` intervention.

*Figure 17.* Layer-wise Integrated Gradients attribution under the ExpeL framework using Qwen3-8B. Prompts are divided into four segments: System Instruction, Condensed Experience, Raw Experience, and Current Trajectory.

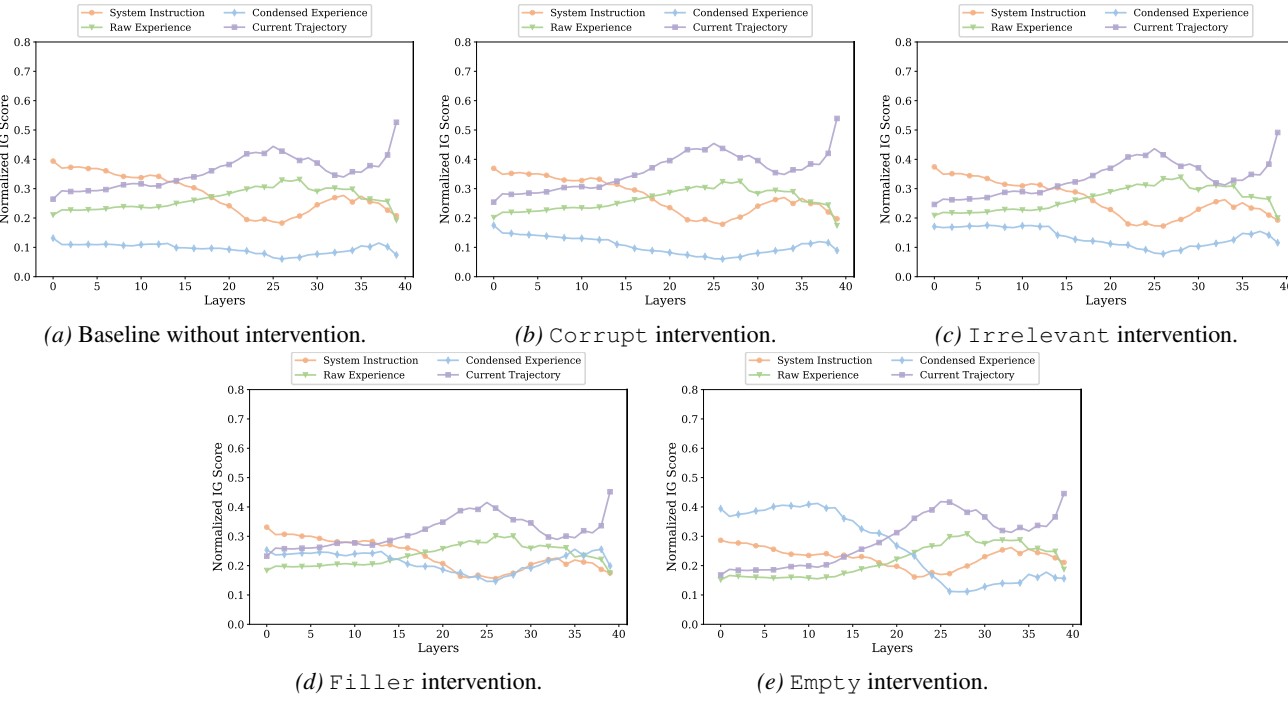

*(a)* Baseline without intervention.     *(b)* `Corrupt` intervention.     *(c)* `Irrelevant` intervention.

*(d)* `Filler` intervention.     *(e)* `Empty` intervention.

*Figure 18.* Layer-wise Integrated Gradients attribution under the ExpeL framework using Qwen3-14B. Prompts are divided into four segments: System Instruction, Condensed Experience, Raw Experience, and Current Trajectory.

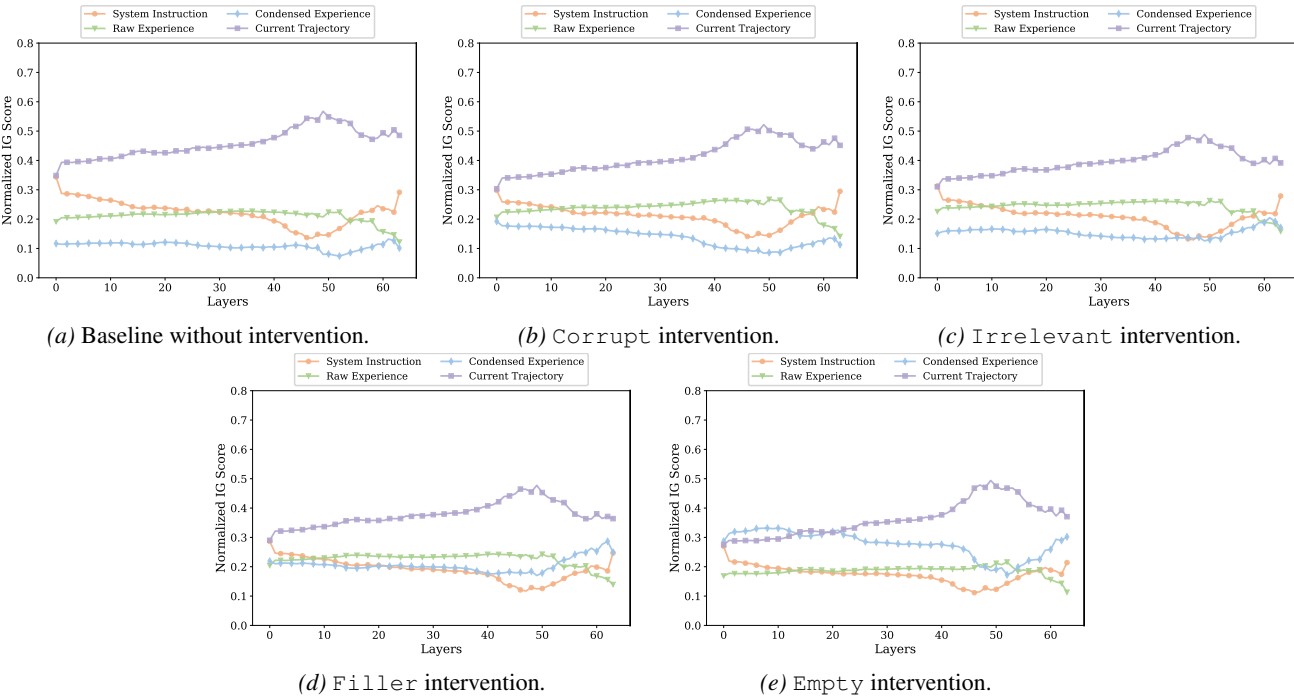

*(a)* Baseline without intervention.     *(b)* `Corrupt` intervention.     *(c)* `Irrelevant` intervention.

*(d)* `Filler` intervention.     *(e)* `Empty` intervention.

*Figure 19.* Layer-wise Integrated Gradients attribution under the ExpeL framework using Qwen3-32B. Prompts are divided into four segments: System Instruction, Condensed Experience, Raw Experience, and Current Trajectory.

*Table 7.* Intervention results on two multi-hop question answering tasks under the ExpeL framework with Qwen3-14B backbone. The evaluation metric is exact match.

| | **2Wiki-MultiHopQA** | **Musique** |
|---|:---:|:---:|
| ExpeL | 60 | 37 |
| *Raw Experience Intervention* | | |
| Empty | 58 | 42 |
| Shuffle | 55 | 42 |
| Irrelevant | 60 | 41 |
| *Condensed Experience Intervention* | | |
| Empty | 60 | 34 |
| Corrupt | 58 | 44 |
| Irrelevant | 59 | 40 |
| Filler | 56 | 44 |

