# OpenReview forum: "Large Language Model Agents Are Not Always Faithful Self-Evolvers"
_ICML.cc/2026/Conference — ICML 2026 regular_

### Official Review · Reviewer_v9gX · 2026-03-12

**Soundness:** 2
**Presentation:** 3
**Significance:** 3
**Originality:** 2
**Overall Recommendation:** 4
**Confidence:** 3

**Summary:**

This paper studies whether self-evolving LLM agents faithfully use retrieved experience. Using controlled interventions on raw trajectories and condensed summaries across multiple agent frameworks, models, and tasks, the paper finds a consistent asymmetry: agents rely strongly on raw experience but often show limited sensitivity to condensed experience. The authors further argue that this gap comes from the weak specificity of condensed memory, model biases toward local context, and task settings where pretrained knowledge is already sufficient.

**Compliance With Llm Reviewing Policy:**

Affirmed.

**Final Justification:**

I am raising my score from 3 (Weak Reject) to 4 (Accept) because the rebuttal substantially strengthened the paper and addressed most of my main concerns. In particular, the authors added variance statistics and significance tests, which meaningfully improve the statistical credibility of the results, and they also expanded the control analysis well beyond the single setting initially discussed, now covering offline single-agent, online single-agent, and multi-agent cases across additional models and tasks. These added experiments make the central empirical claim much more convincing: the raw–condensed faithfulness asymmetry appears robust rather than a single-run artifact or a narrow confound of prompt position or formatting. While I still think some presentation and reproducibility details could be clearer in the final version, the paper’s core contribution is important, the intervention-based methodology is strong and insightful, and the rebuttal gives me enough confidence that the main conclusion is both meaningful and well supported.

**Key Questions For Authors:**

See Weaknesses

**Limitations:**

yes

**Strengths And Weaknesses:**

### Strengths

1. The paper studies a timely and important question for self-evolving LLM agents: whether stored experience is actually used faithfully. The empirical coverage is also strong, spanning multiple frameworks, model backbones, and task settings.

2. The intervention-based methodology is intuitive and leads to a clear takeaway: agents are much more sensitive to raw experience than to condensed experience. This is a useful and practically relevant finding for memory-based agent design.

### Weaknesses

1. **Limited statistical validation.**
  The main results are mostly presented as point estimates. The paper would be stronger with variance reporting, confidence intervals, or significance tests, especially given the noisiness of agentic evaluations.

2. **More detail is needed to rule out intervention confounds.**
  The intervention design is promising, but the setup should be described more carefully to separate semantic effects from possible confounds such as formatting, token length, or prompt position.

3. **Reproducibility and evaluation details could be clearer.**
  Given the number of frameworks and settings involved, the paper would benefit from clearer reporting of evaluation protocols and framework-specific implementation details that may affect the conclusions.

---

> ### Author Rebuttal · Authors · 2026-03-31
>
> We sincerely thank the reviewer for the thoughtful assessment. We are encouraged that the reviewer recognizes both the importance of the question and the value of our broad empirical coverage across frameworks, backbones, and task settings. We also appreciate the acknowledgment that our intervention-based methodology yields a clear and practically relevant takeaway for memory-based agent design.
>
> ---
>
> > **W1: Limited statistical validation**
>
> We agree and have added variance statistics and significance tests (*p* < 0.05), **particularly for small-scale benchmarks and noisy agentic evaluation settings**, including **multi-hop QA (2Wiki, Musique)**, **ALFWorld**, and **WebArena**, using closed-source API models (GPT-4o and Gemini 2.5 Flash). All results are reported as 3-run averages, where the first number in parentheses denotes variance and the second denotes the p-value.
>
> - offline ExpeL with GPT-4o: raw experience interventions on ALFWorld and 2Wiki remain consistently significant, while condensed interventions remain much weaker and often statistically insignificant.
>
> ||ALFWorld|2Wiki|Musique|
> |:---:|:---:|:---:|:---:|
> |ExpeL|74.62(0.56)|57.33(1.33)|44.67(2.33)|
> |Raw Experience Interventions||||
> |Empty |1.74(0.73,0.001)|46.33(2.33,0.001)|42.67(0.33,0.139)|
> |Shuffle|68.16(1.30,0.002)|60.00(4.00,0.134)|41.67(1.33,0.058)|
> |Irrelevant|33.08(0.74,0.001)|55.33(0.33,0.076)|43.00(1.00,0.200)|
> |Condensed Experience Interventions||||
> |Empty|78.61(2.40,0.030)|63.67(2.33,0.006)|42.00(1.00,0.075)|
> |Corrupt|76.62(0.19,0.024)|60.33(1.33,0.033)|41.33(1.33,0.043)|
> |Irrelevant|76.12(3.90,0.320)|61.00(1.00,0.015)|43.67(2.33,0.468)|
> |Filler|74.13(1.30,0.567)|63.33(2.33,0.007)|44.00(0.00,0.529)|
>
> - online ReasoningBank with Gemini 2.5 Flash: condensed interventions across Shopping / CMS / Reddit / Map continue to show limited and unstable effects, despite repeated runs.
>
> ||Shopping|CMS|Reddit|Map|
> |:---:|:---:|:---:|:---:|:---:|
> |ReasoningBank|59.73(1.29)|59.50(0.67)|48.47(1.20)|59.03(3.66)|
> |Condensed Experience Interventions|||||
> |Empty|60.63(0.40,0.314)|64.07(2.82,0.026)|47.80(1.08,0.487)|60.57(7.56,0.477)|
> |Corrupt|59.90(2.01,0.882)|62.80(4.41,0.095)|54.10(2.17,0.008)|56.90(0.81,0.738)|
> |Irrelevant|61.83(0.08,0.077)|59.33(5.82,0.713)|54.40(2.17,0.006)|58.73(5.85,0.875)|
> |Filler|62.90(1.47,0.030)|63.17(7.56,0.097)|53.13(3.97,0.124)|59.93(5.37,0.632)|
>
> These results strengthen our conclusion that the **raw–condensed faithfulness asymmetry is robust**, rather than an artifact of single-run noise. We will include these statistics more clearly in the revision.
>
> ---
>
> > **W2. More detail is needed to rule out intervention confounds**
>
> We fully agree with this suggestion and have conducted additional control experiments to better separate semantic effects from possible confounds such as format and prompt position. Due to rebuttal space, please refer to our response to **Reviewer zWY6, Weakness 2**, where these new results are reported. They further support that the observed raw–condensed gap is not an artifact of intervention design.
>
> ---
>
> > **W3. Reproducibility and evaluation details could be clearer.**
>
> We thank the reviewer for this suggestion. We note that in **`Section 3.1`** we explicitly state that our setups follow the official configurations of each framework (**`Lines 144 – 149`**), and in **`Appendix C`** we provide **detailed hyperparameters and implementation details**. We will further improve clarity in the revision by making these evaluation protocols and settings more explicit and easier to follow.

---

> > ### Author Rebuttal · Reviewer_v9gX · 2026-04-04
> >
> > Thank the author for the careful rebuttal and extra simulations. Several things have been resolved, but key concerns, especially about W2, do not yet appear fully resolved. In particular, the position-control result still shows a significant effect for condensed Empty (p=0.008), suggesting that positional effects may not be fully ruled out, and the new controls are currently demonstrated only on a single benchmark (ALFWorld) with a single model (Qwen3-235B-A22B). Given that the paper’s main strength is its breadth, the confound-ruling argument would be more convincing if similar controls were shown in at least a few additional settings.

---

> > > ### Author Response · Authors · 2026-04-06
> > >
> > > Thank you again for the careful follow-up. We appreciate this important point and have further strengthened the control analysis.
> > >
> > > ---
> > >
> > > > **the position-control result still shows a significant effect for condensed Empty (p=0.008)**
> > >
> > > We would like to clarify that this pattern is actually **fully consistent with our main findings**, including the trend already shown in **Figure 8**: compared with raw experience, agents remain substantially less faithful to condensed experience, while within condensed interventions, Empty consistently has the relatively large effect. In other words, this result does not overturn our conclusion, but rather reproduces the same relative pattern under position swapping. This suggests that **position does not alter the raw–condensed asymmetry**.
> > >
> > > ---
> > >
> > > > **the confound-ruling argument would be more convincing if similar controls were shown in at least a few additional settings**
> > >
> > > To further address the reviewer’s concern, we have now added **additional matched-control experiments** beyond ALFWorld / Qwen3-235B-A22B, covering all major settings in the paper:
> > >
> > > - **ExpeL** on **WebShop** with **GPT-4o** and **Qwen3-235B-A22B**,
> > > - **Dynamic CheatSheet** on **Game of 24** with **GPT-4o**,
> > > - **G-Memory** on **ALFWorld** with **GPT-4o-mini**.
> > >
> > > These additions now span **single-agent offline (ExpeL), single-agent online (Dynamic CheatSheet)**, and **multi-agent (G-Memory)** settings. All results are **averaged over 3 runs**; values in parentheses denote **(variance, p-value)**.
> > >
> > > - **ExpeL** on **WebShop** with **GPT-4o** and **Qwen3-235B-A22B**
> > >
> > > **Position-control:**
> > >
> > > |Webshop|GPT-4o|Qwen3-235B-A22B|
> > > |:---:|:---:|:---:|
> > > |+ExpeL|31.67(0.3)|28.67(0.3)|
> > > |Raw Experience Interventions|||
> > > |+Empty|27.67(4.3,0.070)|24.67(1.3,0.013)|
> > > |+Shuffle|30.00(1.0,0.082)|25.67(2.3,0.062)|
> > > |+Irrelevant|27.33(0.3,0.001)|27.33(0.3,0.047)|
> > > |Condensed Experience Interventions|||
> > > |+Empty|30.67(2.3,0.379)|30.67(0.3,0.013)|
> > > |+Corrupt|31.67(6.3,1.000)|29.00(3.0,0.777)|
> > > |+Irrelevant|32.33(0.3,0.230)|29.33(0.3,0.230)|
> > > |+Filler|31.00(3.0,0.581)|29.00(1.0,0.649)|
> > >
> > > **Format-control:**
> > >
> > > |Webshop|GPT-4o|Qwen3-235B-A22B|
> > > |:---:|:---:|:---:|
> > > |+ExpeL|31.67(0.3)|28.67(0.3)|
> > > |Raw Experience Interventions|||
> > > |+Corrupt|26.33(2.3,0.017)|26.33(4.3,0.185)|
> > >
> > > - **Dynamic CheatSheet** on **Game of 24** with **GPT-4o**
> > >
> > > **Position-control:**
> > >
> > > |GPT-4o|GameOf24|
> > > |:---:|:---:|
> > > |+DC_RS|94.67(0.3)|
> > > |Raw Experience Interventions||
> > > |Empty|36.67(2.3,0.001)|
> > > |Shuffle|56.00(7.0,0.001)|
> > > |Irrelevant|86.33(2.3,0.006)|
> > > |Condensed Experience Interventions||
> > > |Empty|90.67(2.3,0.033)|
> > > |Corrupt|90.33(2.3,0.027)|
> > > |Irrelevant|92.33(1.3,0.053)|
> > > |Filler|92.67(2.3,0.139)|
> > >
> > > **Format-control:**
> > >
> > > |GPT-4o|GameOf24|
> > > |:---:|:---:|
> > > |+ExpeL|94.67(0.3)|
> > > |Raw Experience Interventions||
> > > |Corrupt|63.00(3.0,0.001)|
> > >
> > > - **G-Memory** on **ALFWorld** with **GPT-4o-mini**: Since G-Memory contains **two distinct types of raw experience**, we conduct position-control and format-control experiments for each raw source separately.
> > >
> > > **Position-control:**
> > >
> > > |GPT-4o-mini|ALFWorld|
> > > |:---:|:---:|
> > > |+G-Memory|78.4(2.56)|
> > > |Swap Ref-Raw & Condensed||
> > > |+Empty_ref-raw|41.77(3.86,0.000021)|
> > > |+Shuffle_ref-raw|53.70(2.25,0.000039)|
> > > |+Irrelevant_ref-raw|45.53(2.70,0.000026)|
> > > |+Empty_exe-raw|41.03(8.70,0.000258)|
> > > |+Shuffle_exe-raw|46.27(1.61,0.000018)|
> > > |+Irrelevant_exe-raw|42.03(0.11,0.000475)|
> > > |+Empty_condensed|73.87(0.56,0.024745)|
> > > |+Corrupt_condensed|76.57(3.30,0.273346)|
> > > |+Irrelevant_condensed|72.60(3.00,0.013761)|
> > > |+Filler_condensed|75.87(1.55,0.128050)|
> > > |Swap Exe-Raw & Condensed||
> > > |+Empty_ref-raw|37.57(3.84,0.000050)|
> > > |+Shuffle_ref-raw|42.30(5.25,0.000055)|
> > > |+Irrelevant_ref-raw|35.83(7.78,0.000412)|
> > > |+Empty_exe-raw|45.80(3.42,0.000091)|
> > > |+Shuffle_exe-raw|51.97(1.61,0.000035)|
> > > |+Irrelevant_exe-raw|65.93(1.61,0.000643)|
> > > |+Empty_condensed|73.40(0.50,0.0181)|
> > > |+Corrupt_condensed|75.90(0.50,0.1043)|
> > > |+Irrelevant_condensed|73.37(1.14,0.012080)|
> > > |+Filler_condensed|75.63(1.20,0.0853)|
> > >
> > > **Format-control:**
> > >
> > > |GPT-4o-mini|ALFWorld|
> > > |:---:|:---:|
> > > |+GMemory|78.4(2.56)|
> > > |Ref-Raw Interventions||
> > > |+Corrupt|37.8(2.46,0.000005)|
> > > |Exe-Raw Interventions ||
> > > |+Corrupt|55.5(5.25,0.00014)|
> > >
> > > Across all these cases, the conclusions remain unchanged:
> > >
> > > (1) under **position control**, agents are still **consistently less faithful to condensed experience than to raw experience**;
> > >
> > > (2) under **format control**, when the **same style of intervention** is applied, perturbing **raw experience** still causes **substantially larger performance degradation**.
> > >
> > > ---
> > >
> > > We hope these additional experiments and clarifications help fully address your concern.

---

### Official Review · Reviewer_zWY6 · 2026-03-13

**Soundness:** 3
**Presentation:** 3
**Significance:** 3
**Originality:** 3
**Overall Recommendation:** 6
**Confidence:** 4

**Summary:**

The paper studies whether self-evolving LLM agents actually use their accumulated experience. It distinguishes raw experience from condensed experience, and operationalizes “experience faithfulness” via controlled interventions: if perturbing the provided experience causes substantial behavioral changes, the agent is considered faithful to that experience. The evaluation spans 4 representative frameworks, 10 backbones, and 9 environments/tasks. The main finding is a strong asymmetry: agents are consistently more sensitive to perturbations of raw experience than to perturbations of condensed experience, even in condensed-only settings.

**Compliance With Llm Reviewing Policy:**

Affirmed.

**Final Justification:**

All weaknesses have been fully resolved. I will accordingly update my scores: raising Soundness from 2 to 3, and my overall recommendation from Weak Accept to Strong Accept, as the paper now meets the high standards of ICML.

**Key Questions For Authors:**

1. Can you provide a trajectory-level faithfulness metric, rather than relying primarily on final success rate? Examples include action edit distance, tool-use divergence, state visitation differences, or answer-preserving behavioral changes. If these align with your current claims, I would raise my soundness score.

2. Can you add stricter length-/position-/format-matched controls to rule out confounds from asymmetric intervention strength between raw and condensed memories? If the gap persists under those controls, the main conclusion would be much stronger.

3. Please report variance and statistical significance, especially for WebArena, ALFWorld, and closed-source API models. Would multiple runs, confidence intervals, or significance testing change any of the main conclusions?

4. In Section 5, which of the three proposed causes are intended as explanatory hypotheses, and which do you consider sufficiently supported to warrant causal claims? A clearer claim hierarchy would help.

**Limitations:**

I did not find an explicit Limitations / Broader Impact / societal impact section in the PDF. The paper should at least discuss:
(1) the limitations of operationalizing faithfulness via aggregate performance sensitivity;
(2) potential confounds in the comparability of raw vs. condensed interventions;
(3) possible real-world risks of unfaithful memory use, such as following stale heuristics, mis-executing web actions, or compounding errors in embodied settings.

**Strengths And Weaknesses:**

Strengths

> 1. The paper targets a genuinely important assumption in the current agent literature: improved performance with memory does not necessarily imply faithful use of memory.

> 2. The empirical coverage is broader than many analysis papers: it spans offline vs. online, single-agent vs. multi-agent, and diverse domains including web interaction, embodied tasks, mathematical reasoning, and QA, with broadly consistent trends across diverse methods

> 3. Several diagnostics are genuinely interesting. In ExpeL, removing or perturbing raw trajectories often causes large drops, while condensed-memory perturbations often have much smaller effects. What is more, this paper goes beyond reporting benchmark outcomes by attempting mechanistic explanations via failure taxonomies, integrated-gradients attribution, and task-regime analysis, which makes the work more than a simple evaluation paper.


Weaknesses

> 1. The central operationalization is too coarse for the strength of the claims. The paper effectively uses changes in aggregate success rate under interventions as evidence of faithfulness. But performance sensitivity is not the same as semantic causal reliance: trajectories may change while final success stays constant, and performance drops may partly reflect prompt disruption, context-length effects, or formatting instability rather than genuine semantic dependence. This is useful as a first diagnostic, but too weak to fully support the paper’s stronger “causal grounding” language.

> 2. The intervention families are not fully comparable across raw and condensed memories. Raw memories are long procedural trajectories with temporal structure, whereas condensed memories are short abstractions. Shuffling or replacing raw trajectories may inherently be more destructive than corrupting or filling condensed summaries. Thus part of the reported gap may reflect differences in information density, prompt position, and actionability, not only differences in “faithfulness.”

> 3. The statistical rigor is limited. Many evaluations use only 100 examples, including the multi-hop QA study, and I did not find confidence intervals, standard deviations, p-values, or significance tests in the paper. For noisy settings such as web interaction, closed-model APIs, and online memory updates, point estimates alone are not fully convincing by ICML standards.

> 4. The strongest challenge to the paper’s broad framing comes from its own results. In Table 2 on 2Wiki-MultiHopQA and Musique, neither raw nor condensed interventions produce stable degradation; some perturbed conditions even outperform the original ExpeL setup. This suggests the phenomenon is more task-dependent than the current framing admits.

---

> ### Author Rebuttal · Authors · 2026-03-31
>
> > **W1 & Q1: trajectory-level metrics**
>
> We follow this suggestion by adding trajectory-level faithfulness evaluation. Specifically, under the ExpeL framework, we report **action edit distance** on ALFWorld and **state visitation difference** on WebShop, using GPT-4o and Qwen3-235B-A22B.
>
> - On ALFWorld, raw interventions induce larger action edit distances than condensed ones.
>
> ||GPT-4o|Qwen3-235B|
> |:---:|:---:|:---:|
> |Raw Experience Interventions|||
> |Empty |12.31|15.90|
> |Shuffle |3.19|3.86|
> |Irrelevant|8.35|9.46|
> |Condensed Experience Interventions|||
> |Empty|2.35|3.81|
> |Corrupt|1.79|2.61|
> |Irrelevant|2.41|3.15|
> |Filler|1.65|3.37|
>
> - On WebShop, we measure state visitation difference as the total variation distance between the normalized webpage-visit distributions of the original and intervened trajectories (range: 0–1, where 0 indicates identical visitation and 1 indicates no overlap). Again, raw interventions produce much larger deviations than condensed ones.
>
> ||Qwen3-235B|GPT-4o|
> |:---:|:---:|:---:|
> |Raw Experience Interventions|||
> |Empty|0.66|0.68|
> |Shuffle|0.34|0.39|
> |Irrelevant|0.32|0.32|
> |Condensed Experience Interventions|||
> |Empty|0.23|0.22|
> |Corrupt|0.17|0.22|
> |Irrelevant|0.22|0.28|
> |Filler|0.20|0.22|
>
> These results are highly **consistent with our outcome-level findings**: raw experience perturbations induce substantially stronger behavioral deviations than condensed ones. We will incorporate these trajectory-level analyses in the revision.
>
> ---
>
> > **W2 & Q2. Raw vs. condensed interventions are not fully comparable; can you add stricter matched controls?**
>
> In fact, the **Filler** perturbation (Table 3) to condensed experience is arguably the **most destructive**, as it completely removes semantic content and replaces it with placeholders. Despite this, model performance remains largely unchanged, suggesting that the weak sensitivity to condensed experience is not due to insufficient perturbation strength.
>
> To further rule out confounds from asymmetric intervention, we follow your suggestion and introduce **stricter matched controls** on the ExpeL framework (which contains both raw and condensed experience), using Qwen3-235B-A22B (3 runs; we report variance and significance with *p* < 0.05).
>
> - Position control. We swap the positions of raw and condensed experience before applying perturbations. The results show that **raw interventions still lead to substantial degradation**, while **condensed interventions remain weak**.
>
> |Qwen3-235B-A22B|ALFWorld|
> |:---:|:---:|
> |ExpeL|78.85(4.64)|
> |Raw Experience Interventions||
> |Empty |4.23(0.74,0.001)|
> |Shuffle |75.37(7.26,0.159)|
> |Irrelevant |42.04(5.77,0.001)|
> |Condensed Experience Interventions||
> |Empty |70.65(3.53,0.008)|
> |Corrupt |77.12(1.30,0.304)|
> |Irrelevant |75.62(2.42,0.110)|
> |Filler |73.38(5.76,0.043)|
>
> - Format control. We apply the same corruption-style perturbation to raw experience, matching the format used for condensed summaries. Even under this aligned perturbation, raw performance drops significantly.
>
> |Qwen3-235B-A22B|ALFWorld|
> |:---:|:---:|
> |+ExpeL|78.85(4.64)|
> |Raw Experience Interventions||
> |Corrupt |60.20(4.64, 0.001)|
>
> Overall, these results confirm that the observed gap is **not driven by differences in intervention strength, position, or format**, but reflects a more fundamental asymmetry in how current agents utilize raw and condensed experience. We will include these analyses in the revision.
>
> ---
>
> > **W3 & Q3. report variance / significance.**
>
> We have added variance and significance tests, especially WebArena, ALFWorld, and closed-source API models. Across repeated runs, the **main conclusions remain stable**. Due to rebuttal space, please refer to our response to **Reviewer v9gX, Weakness 1**.
>
> ---
>
> > **W4 & Q4. The phenomenon appears task-dependent; what in Section 5 is causal claim vs. explanatory hypothesis?**
>
> We agree that the phenomenon is not uniform across all tasks, and we have already acknowledged and analyzed this explicitly in **`Section 5.3`** and the **`Impact Statement`**. As organized in Section 5, we trace the cause of unfaithfulness through the **three core components of a self-evolving system**: the **experience itself** (Section 5.1), the **backbone model** that processes it (Section 5.2), and the **task environment** in which the agent operates (Section 5.3).
>
> Within this hierarchy, **Section 5.1** and **Section 5.3** are directly supported by controlled experiments, and are therefore intended as **empirically supported findings**. In contrast, **Section 5.2** serves as a **mechanistic explanation**, where attribution analysis provides supporting evidence rather than a definitive causal claim. We will make this claim hierarchy more explicit in the revision.
>
> ---
>
> > **Limitations / Broader Impact.**
>
> We already discuss broader implications in the **`Impact Statement`**. We will incorporate a dedicated Limitations section in the revision and expand the discussion of real-world risks.

---

> > ### Author Rebuttal · Reviewer_zWY6 · 2026-04-04
> >
> > Thank you for the detailed and thorough rebuttal that addresses all my concerns. I am particularly impressed by the substantial new experiments you have conducted to resolve the raised issues.
> >
> > All weaknesses have been fully resolved. I will accordingly update my scores: raising Soundness from 2 to 3, and my overall recommendation from Weak Accept to Strong Accept, as the paper now meets the high standards of ICML.

---

> > > ### Author Response · Authors · 2026-04-05
> > >
> > > Thank you very much for the thoughtful follow-up and for the encouraging feedback. We truly appreciate your constructive suggestions, which greatly helped improve the paper and strengthen it toward the high standards of ICML.
> > >
> > > May we kindly check one small detail: in your comment, you mentioned **updating the overall recommendation to Strong Accept (6)**, while the currently displayed score seems to be Accept (5). We were just wondering whether the score is mis-selected. Thanks so much for your considerations.

---

### Official Review · Reviewer_cAW4 · 2026-03-13

**Soundness:** 3
**Presentation:** 3
**Significance:** 2
**Originality:** 2
**Overall Recommendation:** 3
**Confidence:** 3

**Summary:**

This paper investigates whether self-evolving LLM agents faithfully utilize the experience they accumulate through interaction. The authors distinguish between raw experience and condensed experience, and design causal intervention experiments to probe whether agents actually rely on retrieved experience or merely appear to benefit from it. Experiments span four agent frameworks, 10 LLMs, and 9 benchmarks. The key finding is that while raw experience is faithfully utilized and its removal causes substantial performance drops, condensed experience is largely ignored. Agents show minimal behavioral change when condensed summaries are perturbed. Additional analyses including failure case studies support these conclusions.

**Compliance With Llm Reviewing Policy:**

Affirmed.

**Final Justification:**

After carefully considering the authors' rebuttal, my main concerns remain insufficiently addressed, and I am unable to recommend acceptance at this time.

**Key Questions For Authors:**

In several experiments, removing condensed experience actually improves performance over the baseline. This suggests condensed experience may be actively harmful in some settings, which is a different and arguably more concerning finding than mere unfaithfulness. It deserves more discussion.

For other questions, please refer to the weaknesses above.

**Limitations:**

yes

**Strengths And Weaknesses:**

**Strengths:**

1. The research question is well-motivated and practically important.
2. The experimental design is thorough.
3. Breadth of evaluation is impressive.
4. The failure case analysis is valuable, providing concrete qualitative evidence of how agents mishandle condensed experience.

**Weaknesses:**

1. The core finding that condensed experience has minimal impact may partly reflect the quality of the condensation process rather than a fundamental limitation of LLM agents. If the summaries generated by these systems are genuinely low-quality or redundant with the model's parametric knowledge, the finding is less surprising. The paper does not control for condensation quality or test with human-written high-quality condensed experience.

2. The paper frames this as a faithfulness problem, but the terminology is somewhat overloaded. "Faithfulness" in the LLM literature often refers to whether explanations match actual reasoning. Here it means whether behavioral outputs change when experience is perturbed. This conflation could cause confusion, and the connection to the broader faithfulness literature is not well articulated.

3. The attribution analysis via integrated gradients is interesting but limited in interpretive power.

4. The practical takeaway is somewhat thin. The paper identifies a gap but the "design takeaways" are fairly generic. These do not constitute concrete solutions or new methods.

5. Some experimental choices are not fully justified. For instance, the benchmark s9ze is relatively small, and confidence intervals or significance tests are largely absent from the main results.

---

> ### Author Rebuttal · Authors · 2026-03-31
>
> We thank the reviewer for the thorough and constructive feedback. We are glad the reviewer found the research question well-motivated, the experimental design thorough, the breadth of evaluation impressive, and the failure case analysis valuable. We address each concern below.
>
> ---
>
> > **W1: Condensation quality not controlled.**
>
> Our findings suggest that the issue is **not solely attributable to low-quality summaries**. As shown in our model-scaling analysis (**`Section 4.4, Figure 6`**), although increasing model capacity substantially improves overall performance—and larger models should in principle generate **higher-quality condensed experience** than smaller ones, since the condensed experience are extracted by the models themselves—both small and large models remain consistently insensitive to condensed interventions. This suggests that the observed unfaithfulness reflects a more fundamental limitation in how current agents utilize condensed experience, rather than merely the quality of the summaries themselves.
>
> ---
>
> > **W2: Faithfulness terminology is overloaded.**
>
> In fact, both previous explanation faithfulness and our behavioral faithfulness fundamentally study the same question: **to what extent model outputs (reasoning outcome or behavioral actions) are influenced by their provided context**. In this sense, they are conceptually aligned but instantiated in different settings. We connect our work to the broader faithfulness literature in Related Work (`Lines 409 - 422`), tracing the progression from in-context learning to CoT faithfulness, and positioning our setting as a natural extension to **dynamic, agentic experience use**. We will further clarify this distinction to avoid ambiguity.
>
> ---
>
> > **W3: IG analysis has limited interpretive power.**
>
> We use IG as a **mechanistic probe** to explain *why* agents fail to faithfully utilize condensed experience, rather than as standalone evidence. As shown in `Section 5.2` and `Appendix E.5`, experiments across **Qwen3 models from 1.7B to 32B exhibit consistent attribution patterns**, providing stable support for our interpretation. We will clarify this role in the paper and avoid overstating its causal claims.
>
> ---
>
> > **W4: The practical takeaway is somewhat thin.**
>
> We appreciate this suggestion and are glad to share a new result directly motivated by our second design insight (`Lines 468–479`) in the Impact Statement: rethinking the **timing of experience use**. Instead of retrieving condensed experience in a fixed manner at the beginning of each task (as in standard ReasoningBank), we implement a simple variant (ReasoningBank_Dynamic) that allows the model to retrieve experience only when it judges such guidance to be necessary. Importantly, this is achieved **purely through prompt engineering**—by reminding the model to retrieve experience on demand—without introducing any additional training.
>
> We report preliminary results on Qwen3-32B over ALFWorld and FEVER (3-run average; values in parentheses denote performance change under intervention). Compared with standard ReasoningBank, the dynamic variant not only achieves **higher task performance**, but also shows **larger drops under condensed perturbations**, indicating **stronger faithfulness** to condensed experience. This reveals the key limitation of current frameworks regarding the use of experience. We believe this is an important direction for future work.
>
> - ALFWorld
>
> |Qwen3-32B|ReasoningBank|ReasoningBank_Dynamic|
> |:---:|:---:|:---:|
> |Original|69.4|**73.4**|
> |Empty|68.4(-1.0)|69.6 (**-3.8**) |
> |Corrupt|69.2(-0.2)|70.1(**-3.3**)|
> |Irrelevant|68.4(-1.0)|67.7(**-5.7**)|
> |Filler|68.2(-1.2)|71.1(**-2.3**)|
>
> - FEVER
>
> |Qwen3-32B|ReasoningBank|ReasoningBank_Dynamic|
> |:---:|:---:|:---:|
> |Original|64.7|**68.0**|
> |Empty|65.3(+0.6)|64.3(**-3.7**)|
> |Corrupt|68.0(+3.3)|64.7(**-3.3**)|
> |Irrelevant|63.7(-1.0)|64.7(**-3.3**)|
> |Filler|68.0(+3.3)|62.3(**-5.7**)|
>
> ---
>
> > **W5: Missing significance tests**
>
> Due to the rebuttal character limit, please refer to our response to **Reviewer v9gX, Weakness 1**, where we report additional variance statistics and significance tests. These results support the **stability and robustness** of our main findings. We will incorporate this more clearly in the revised paper.
>
> ---
>
> > **Q1: Condensed experience sometimes hurts performance.**
>
> We fully agree. In fact, we already discuss this phenomenon explicitly in **`Section 5.1`**, where we show that condensed experience can be actively harmful by inducing distraction, incorrect priors, and premature inference. We also reflect this in the **`Impact Statement`**, where we emphasize the corresponding design insight. We will make this point more prominent in the revised paper.

---

> > ### Author Rebuttal · Reviewer_cAW4 · 2026-04-04
> >
> > I thank the authors for their detailed rebuttal. However, I believe fully addressing the raised concerns would require substantial additional experiments.

---

> > > ### Author Response · Authors · 2026-04-07
> > >
> > > Thank you again for the careful follow-up. We appreciate the reviewer’s continued engagement and understand the concern that fully resolving some issues may require stronger empirical support. In particular, for the concerns that involve additional experiments (W1, W4, and W5), we have conducted further analyses during the rebuttal period to strengthen the empirical evidence.
> > >
> > > ---
> > >
> > > > **W1: Condensation quality not controlled.**
> > >
> > > We further evaluate **condensed-experience interventions** under the ExpeL framework on HotPotQA, using **six Qwen3 models ranging from 1.7B to 32B**. All results are reported as 3-run averages, with (variance, p-value) in parentheses. The results remain highly consistent: regardless of model scale—which also roughly reflects the quality of self-generated condensed experience—agents continue to show **very limited sensitivity to condensed interventions**. Together with our existing model-scaling results in **`Section 4.4 / Figure 6`**, this provides further evidence that the issue is **not solely due to weak summary quality**, but reflects a broader limitation in how current agents utilize condensed experience.
> > >
> > >
> > > |HotPotQA|Qwen3-1.7B|Qwen3-4B|Qwen3-8B|Qwen3-14B|Qwen3-30B-A3B|Qwen3-32B|
> > > |:---:|:---:|:---:|:---:|:---:|:---:|:---:|
> > > |+ExpeL|3.00(0.0)|6.00(0.0)|17.33(0.3)|39.33(2.3)|38.67(0.3)|41.67(2.3)|
> > > |Empty|1.00(0.0,1.000)|5.33(0.3,0.002)|6.00(0.0,0.001)|40.67(0.3,0.267)|41.00(0.0,0.020)|44.00(0.0,0.118)|
> > > |Corrupt|2.67(2.3,0.742)|7.33(2.3,0.269)|22.67(2.3,0.017)|41.00(0.0,0.199)|35.33(4.3,0.100)|42.67(0.3,0.379)|
> > > |Irrelevant|3.33(0.3,0.423)|5.33(2.3,0.017)|16.33(2.3,0.378)|38.67(2.3,0.621)|40.33(10.3,0.465)|42.33(1.3,0.581)|
> > > |Filler|2.67(1.3,0.667)|6.00(4.0,0.035)|15.67(2.3,0.191)|38.33(1.3,0.420)|40.00(1.0,0.134)|41.33(0.3,0.751)|
> > >
> > > ---
> > >
> > > > **W4: The practical takeaway is somewhat thin.**
> > >
> > > We also add further evaluation of ReasoningBank_Dynamic on **multi-hop QA benchmarks (HotPotQA and 2Wiki)**, again using 3-run averages. The results continue to show that **ReasoningBank_Dynamic is more faithful** than the original ReasoningBank. Importantly, this also speaks to your **Q1**: this more adaptive way of using experience is not only **more faithful**, but also **more effective**, without sacrificing performance. We believe this provides a concrete design insight for future work on **better experience utilization strategies**.
> > >
> > > - HotpotQA
> > >
> > > |Qwen3-32B|ReasoningBank|ReasoningBank_Dynamic|
> > > |:---:|:---:|:---:|
> > > |Original|42.0|**45.0**|
> > > |Empty|41.3(-0.7)|42.3 (**-2.7**)|
> > > |Corrupt|42.0(0.0)|43.0(**-2.0**)|
> > > |Irrelevant|41.0(-1.0)|40.3(**-4.7**)|
> > > |Filler|42.7(+0.7)|43.0(**-2.0**)|
> > >
> > > - 2Wiki
> > >
> > > |Qwen3-32B|ReasoningBank|ReasoningBank_Dynamic|
> > > |:---:|:---:|:---:|
> > > |Original|62.0|**64.3**|
> > > |Empty|60.3(-1.7)|62.0(**-2.3**)|
> > > |Corrupt|61.0(-1.0)|61.3(**-3.0**)|
> > > |Irrelevant|60.3(-1.7)|59.3(**-5.0**)|
> > > |Filler|61.7(-0.3)|60.0(**-4.3**)|
> > >
> > > ---
> > >
> > > > **W5: Missing significance tests**
> > >
> > > In addition to the significance analyses reported in our response to **Reviewer v9gX, Weakness 1**, we further add **statistical significance tests** under the **multi-agent G-Memory** framework, using GPT-4o-mini on ALFWorld. With this addition, our significance analysis now covers the majority of the paper’s settings, including **single-agent offline (ExpeL)**, **single-agent online (ReasoningBank)**, and **multi-agent (G-Memory)** scenarios. **Together, these results further support the stability and robustness of our conclusions.**
> > >
> > > |GPT-4o-mini|ALFWorld|
> > > |:---:|:---:|
> > > |+G-Memory|71.6(0.56)|
> > > |Ref-Raw Exp. Intervention||
> > > |Empty|51.5(2.25,0.000275)|
> > > |Shuffle|56.2(2.41,0.0007)|
> > > |Irrelevant|52.0(2.29,0.0003)|
> > > |Exe-Raw Exp. Intervention||
> > > |Empty|56.5(3.00,0.0013)|
> > > |Shuffle|65.2(4.56,0.0243)|
> > > |Irrelevant|51.7(3.46,0.0009)|
> > > |Condensed Exp. Intervention||
> > > |Empty|72.1(2.46,0.6543)|
> > > |Corrupt|69.7(5.62,0.2979)|
> > > |Irrelevant|70.6(1.26,0.2786)|
> > > |Filler|70.4(1.26,0.1902)|
> > >
> > > ---
> > >
> > > We hope these additional experiments help address your remaining concerns.

---

### Official Review · Reviewer_iRf6 · 2026-03-16

**Soundness:** 4
**Presentation:** 3
**Significance:** 4
**Originality:** 3
**Overall Recommendation:** 6
**Confidence:** 4

**Summary:**

The paper investigates whether self-evolving agents actually give weight to learned heuristics and past raw traces. The authors find that causal interventions to the former are basically ineffectual, while causal interventions to the latter have a significant impact. The authors then investigate potential reasons for this asymmetry.

**Compliance With Llm Reviewing Policy:**

Affirmed.

**Key Questions For Authors:**

N/A

**Limitations:**

Yes

**Strengths And Weaknesses:**

Strengths:
1. The paper focuses on a problem that is both relevant and interesting, i.e. "do self-evolving agents actually benefit from what they learn?". The results seem to challenge the traditional consensus that the learned heuristics are useful, since applying perturbations to the latter has a very small effect. This is very interesting.
2. The evaluation is in-depth and involves a mix of causal interventions and interpretability tools, both of which support the authors' claims
3. The paper is relatively well-written and flows well
4. I particularly appreciate the scale-based and IG-based analyses, as they already answer questions I would've asked

Weaknesses:
1. I would've appreciated the use of stronger models (e.g. GPT-5.2, Gemini 3, Claude 4.6 in their non-mini forms) alongside weaker ones, as that would make the results more general. I would be willing to raise my score to a Strong Accept should the authors be willing to add these results (though I understand that running such experiments would be expensive)

Edit: The authors have added the experiments I requested. I am therefore raising my score to Strong Accept.

---

> ### Author Rebuttal · Authors · 2026-03-31
>
> We sincerely thank the reviewer for the very positive assessment. We are especially encouraged that the reviewer finds the core question important and interesting, and appreciates both the depth of evaluation and the scale-based / IG-based analyses. We are also glad that the overall writing and flow were found clear.
>
> ---
>
> > **W1. Stronger frontier models would improve generality**
>
> We fully agree and, following this suggestion, we have added additional results with stronger frontier models, within our computational budget. Specifically, we supplement: ExpeL with Claude-4.6-Sonnet, ReasoningBank with Gemini-3-Pro, and G-Memory with GPT-5.2.
>
> These additions cover the three major self-evolving paradigms—offline, online, and multi-agent—across multiple benchmarks including ALFWorld, HotpotQA, WebShop, WebArena, and FEVER.
>
> - **ExpeL with Claude-4.6-Sonnet**
>
> |Claude-4.6-Sonnet|HotpotQA|ALFWorld|WebShop|
> |:---:|:---:|:---:|:---:|
> |+ExpeL|53|92.54|26|
> |Raw Experience Interventions|||
> |Empty |56|66.42|20|
> |Shuffle|58|92.54|20|
> |Irrelevant|58|87.31|16|
> |Condensed Experience Interventions||||
> |Empty |47|83.58|34|
> |Corrupt |56|90.3|23|
> |Irrelevant |49|91.79|29|
> |Filler|50|88.81|26|
>
> - **ReasoningBank with Gemini-3-Pro**
>
> |Gemini-3-pro|Shopping|CMS|Reddit|Map|
> |:---:|:---:|:---:|:---:|:---:|
> |ReasoningBank|64.7|69.7|54.3|67.0|
> |Condensed Experience Interventions|||||
> |Empty|57.2|63.7|60.4|66.1|
> |Corrupt|61.5|63.2|65.1|68.8|
> |Irrelevant|58.3|61.5|64.2|70.6|
> |Filler|61.5|65.3|59.4|72.5|
>
> - **G-Memory with GPT-5.2**
>
> |GPT-5.2|Fever|ALFWorld|
> |:---:|:---:|:---:|
> |+G-Memory|64|75.4|
> |Ref-Raw Experience Intervention|||
> |Empty|60|51.5|
> |Shuffle|71|67.9|
> |Irrelevant|62|47.0|
> |Exe-Raw Experience Intervention|||
> |Empty|66|55.2|
> |Shuffle|68|47.0|
> |Irrelevant|62|52.2|
> |Condensed Experience Intervention|||
> |Empty|69|85.1|
> |Corrupt|70|82.1|
> |Irrelevant|66|76.9|
> |Filler|71|83.6|
>
> Importantly, the results exhibit the **same overall pattern** as our main findings: on knowledge-intensive QA tasks such as HotpotQA and FEVER, agents remain largely insensitive to perturbations of both raw and condensed experience, while on more interactive agentic tasks, they continue to rely much more faithfully on raw experience than on condensed experience. This suggests that the current limitation in experience utilization persists **even in stronger frontier models**, rather than being confined to weaker backbones.
>
> We thank the reviewer again for this valuable suggestion, which helped us further strengthen the generality of the paper.

---

> > ### Author Rebuttal · Reviewer_iRf6 · 2026-03-31
> >
> > The authors have performed the experiments I've requested, and none of the weaknesses identified by the other reviewers detract, in my opinion, from the core contribution (i.e. "summaries are relatively useless"). I am raising my score to Strong Accept.

---

> > > ### Author Response · Authors · 2026-04-01
> > >
> > > Thank you very much for the thoughtful follow-up and for raising your score to Strong Accept. We are truly grateful for your encouragement and are glad that the additional experiments helped strengthen the paper. We deeply appreciate your time and constructive feedback.

---

### Decision · Program_Chairs · 2026-04-30

**Decision:**

Accept (regular)

**Comment:**

This paper provides the first systematic, causally grounded investigation of faithfulness in self‑evolving LLM agents, revealing a critical and robust asymmetry between raw and condensed experience.

The findings will likely influence future agent designs toward more reliable experience integration.

The authors did a good job at rebuttal but also should include the rebuttal experiments (trajectory metrics, matched controls, significance tests, dynamic variant) in the final version and clarify the claim hierarchy regarding task dependency.

about reviews: iRf6 and zWY6 made excellent, high‑impact suggestions (frontier models, trajectory metrics, matched controls) that materially improved the paper. Rev v9gX was thorough and correct about statistical validation and confounds, though slightly over‑cautious; nonetheless; Rev cAW4 raised some valid points (terminology, practical takeaways), particularly the scaling argument and the dynamic retrieval experiment. The demand for additional experiments seems out of scope.

I think this paper has a cute observation about two different kinds of experience that affect LLM agents, and passes the bar of ICML. I vote acceptance.